# Systematic Outliers in Large Language Models

**Yongqi An**[1,2], **Xu Zhao**[1,4], **Tao Yu**[1,2], **Ming Tang**[1,2], **Jinqiao Wang**[1,2,3,4,*]

[1]Foundation Model Research Center, Institute of Automation, Chinese Academy of Sciences, Beijing, China
[2]School of artificial intelligence, University of Chinese Academy of Sciences, Beijing, China
[3]Wuhan AI Research, Wuhan, China, [4]Objecteye Inc., Beijing, China
{yongqi.an, xu.zhao, tangm, jqwang}@nlpr.ia.ac.cn
yutao2022@ia.ac.cn

## Abstract

Outliers have been widely observed in Large Language Models (LLMs), significantly impacting model performance and posing challenges for model compression. Understanding the functionality and formation mechanisms of these outliers is critically important. Existing works, however, largely focus on reducing the impact of outliers from an algorithmic perspective, lacking an in-depth investigation into their causes and roles. In this work, we provide a detailed analysis of the formation process, underlying causes, and functions of outliers in LLMs. We define and categorize three types of outliers—activation outliers, weight outliers, and attention outliers—and analyze their distributions across different dimensions, uncovering inherent connections between their occurrences and their ultimate influence on the attention mechanism. Based on these observations, we hypothesize and explore the mechanisms by which these outliers arise and function, demonstrating through theoretical derivations and experiments that they emerge due to the self-attention mechanism's softmax operation. These outliers act as implicit context-aware scaling factors within the attention mechanism. As these outliers stem from systematic influences, we term them systematic outliers. Our study not only enhances the understanding of Transformer-based LLMs but also shows that structurally eliminating outliers can accelerate convergence and improve model compression. The code is avilable at `https://github.com/an-yongqi/systematic-outliers`.

## 1 Introduction

Large Language Models (LLMs) have recently demonstrated remarkable capabilities (Brown, 2020; Achiam et al., 2023; Touvron et al., 2023a), making them a central topic of research across various domains. Numerous studies have uncovered intriguing phenomena within these models (Dettmers et al., 2022; Xiao et al., 2023b; Sun et al., 2024), which are crucial for advancing the understanding and application of LLMs. Among these phenomena, the presence of outliers, which are values that deviate significantly from the average of their distribution, has garnered considerable attention (Zhang et al., 2024; Kovaleva et al., 2021; Paglieri et al., 2024).

However, research on outliers in LLMs predominantly emphasizes mitigating their impact through algorithmic techniques, often neglecting a thorough exploration of their underlying causes and functional roles. This narrow focus results in two key shortcomings: first, it limits our understanding of why outliers occur and how they influence model behavior (Yin et al., 2023; Xiao et al., 2023a; Hooper et al., 2024); and second, it overlooks the interrelationships between different types of outliers, treating them in isolation rather than as part of a comprehensive, systematic framework (Zhang et al., 2024; Sun et al., 2024; Liao & Monz, 2024). These gaps hinder a deeper understanding of the mechanisms underlying LLMs and constrain opportunities for more effective optimization and broader applications.

To address these gaps, we systematically analyze outliers in LLMs, focusing on their formation, distribution, and roles within the models. We begin by defining and categorizing three types of outliers:

---

[*]Corresponding Author

*activation outliers, weight outliers, and attention outliers.* By examining their distributions across various dimensions, we uncover inherent connections between their occurrences and demonstrate how these outliers collectively influence the attention mechanism. Building on these findings, we propose a hypothesis regarding the formation mechanisms and functions of these outliers, supported by theoretical derivations and experiments. Specifically, we show that these outliers emerge as a result of the self-attention mechanism's softmax operation and act as implicit, context-aware scaling factors in the attention mechanism. Furthermore, our experiments show that structurally eliminating these outliers can accelerate convergence and enhance model compression, providing new insights for future model optimization and design.

Our main contributions are:

- We define and categorize outliers in LLMs into three types—*activation, weight, and attention outliers*—and uncover their systematic relationships.'
- We reveal that these outliers emerge from the self-attention mechanism's softmax operation and function as implicit, context-aware scaling factors.
- We demonstrate that eliminating outliers accelerates convergence and enhances model compression, offering insights for optimizing LLMs.

## 2 RELATED WORK

**Outliers in Large Language Models.** Outliers in LLMs refer to values that deviate significantly from the average of their distribution (Dettmers et al., 2022). Studies have documented various types of outliers in weights, activations, and attention scores, highlighting their presence and impact. For instance, Dettmers et al. (2022) identified activation outliers and proposed quantization techniques to mitigate their effects. Sun et al. (2024) explored the role of large activations as biases in the attention mechanism. Similarly, Zhang et al. (2024) analyzed weight outliers in LayerNorm layers, demonstrating their importance for maintaining language modeling capabilities in models like GPT-2 and LLaMA2-13B. Additionally, Xiao et al. (2023b) introduced the concept of the "Attention Sink," which occurs when a few keys dominate attention scores.

While previous studies recognize the presence of outliers, they typically focus on specific cases or task-specific solutions like quantization and pruning. In contrast, our work provides a systematic categorization of outliers—*activation, weight, and attention outliers*—and reveals their interconnections and collective influence on the attention mechanism in LLMs.

**The Impact of Outliers on Model Performance and Compression.** Outliers significantly affect both the performance and efficiency of LLMs. Previous research has shown that removing outliers without proper handling can severely degrade performance (Puccetti et al., 2021; Kovaleva et al., 2021; Zhang et al., 2024). In quantization, outliers amplify rounding and clipping errors, leading to substantial quantization losses (Wei et al., 2022; Nrusimha et al., 2024; Lin et al., 2024; Liu et al., 2024). Similarly, magnitude-based pruning strategies face challenges in maintaining model performance when outliers are present (Sun et al., 2023). Yin et al. (2023) observed that outliers correlate strongly with layer sparsity, further complicating pruning approaches. Moreover, in KV cache compression, Xiao et al. (2023b) found that attention score outliers associated with specific tokens play a critical role in preserving context.

Despite extensive research on the adverse effects of outliers, their formation mechanisms and functional roles remain largely unexplored. Existing studies focus on mitigating their impact but lack a systematic investigation of their origins. In contrast, our work explores the formation of outliers within the self-attention mechanism, revealing their role as implicit, context-aware scaling factors and proposing structural solutions to enhance model convergence and compression efficiency.

## 3 SYSTEMATIC OUTLIERS: DEFINITION, EXISTENCE, AND LOCALIZATION

**Definition of Outliers in LLMs.** Outliers in LLMs are values that deviate significantly from the average of their distribution, often surpassing a threshold $\tau$. From Figure 1, we observe that three distinct types of outliers—*activation outliers, weight outliers, and attention outliers*—appear systematically in four key locations within LLaMA2-7B. The specific positions of these outliers are

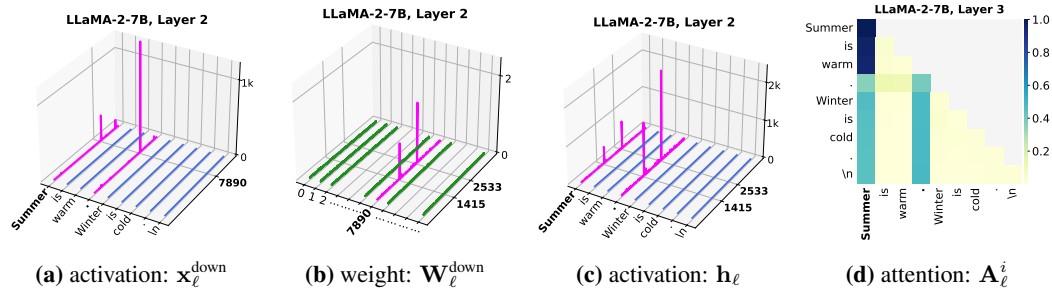

**(a)** activation: $\mathbf{x}_\ell^{\text{down}}$     **(b)** weight: $\mathbf{W}_\ell^{\text{down}}$     **(c)** activation: $\mathbf{h}_\ell$     **(d)** attention: $\mathbf{A}_\ell^i$

**Figure 1:** Systematic outliers in LLaMA2-7B. Outliers are identified in four locations: activations (layer outputs $\mathbf{h}_\ell$ and down-projection inputs $\mathbf{x}_\ell^{\text{down}}$), weights (down-projection matrices $\mathbf{W}_\ell^{\text{down}}$), and attention (attention weights $\mathbf{A}_\ell^i$).

further summarized in Figure 2. To formalize this, we define outliers mathematically for each type as follows:

- **Activation Outliers:** For layer outputs $\mathbf{h}_\ell \in \mathbb{R}^{B \times H}$ (batch size $B$ and hidden dimension $H$), the set of activation outliers $\mathcal{O}_{\text{activation}}$ is:

$$\mathcal{O}_{\text{activation}} = \{(i,j) \mid |h_{i,j}| > \tau \cdot \mu_h\},$$

  where $\mu_h = \frac{1}{B \cdot H} \sum_{i,j} |h_{i,j}|$ is the mean absolute value of $\mathbf{h}_\ell$. Additionally, for down-projection inputs $\mathbf{x}_\ell^{\text{down}} \in \mathbb{R}^{B \times H}$, activation outliers are defined as:

$$\mathcal{O}_{\text{activation-down}} = \{(i,j) \mid |x_{i,j}^{\text{down}}| > \tau \cdot \mu_{x^{\text{down}}}\},$$

  where $\mu_{x^{\text{down}}} = \frac{1}{B \cdot H} \sum_{i,j} |x_{i,j}^{\text{down}}|$.

- **Weight Outliers:** For projection weights $\mathbf{W} \in \mathbb{R}^{O \times I}$ (output dimension $O$, input dimension $I$), weight outliers $\mathcal{O}_{\text{weight}}$ are defined as:

$$\mathcal{O}_{\text{weight}} = \{(i,j) \mid |w_{i,j}| > \tau \cdot \mu_{w_i}\},$$

  where $\mu_{w_i} = \frac{1}{I} \sum_j |w_{i,j}|$ is the row-wise mean absolute value of $\mathbf{W}$.

- **Attention Outliers:** For cumulative attention scores $\mathbf{A} \in \mathbb{R}^{L \times L}$ (sequence length $L$), attention outliers $\mathcal{O}_{\text{attention}}$ are:

$$\mathcal{O}_{\text{attention}} = \{j \mid \hat{A}_j > \tau \cdot \mu_A\},$$

  where $\hat{A}_j = \sum_{i=1}^{L} A_{i,j}$ is the cumulative attention contribution for token $j$, and $\mu_A = \frac{1}{L} \sum_j \hat{A}_j$.

These definitions provide a unified framework to identify and analyze outliers across LLM components. We empirically set $\tau = 1000$ in our experiments to isolate extreme deviations.

**Existence of Outliers in LLMs.** Systematic outliers consistently emerge in LLMs across various components and architectures. Figure 1 highlights their presence in LLaMA2-7B (Touvron et al., 2023b), where we observe abnormally large values in activations, weights, and attention scores at specific indices. These patterns indicate the regular appearance of systematic outliers at critical positions within the model.

This phenomenon extends beyond LLaMA2-7B to a wide range of LLMs, spanning diverse model sizes and families. Our analysis confirms that systematic outliers are a common feature across pretrained and fine-tuned LLMs. For additional examples from other architectures, refer to Appendix A. To deepen our understanding, we next analyze their distributions across various dimensions, providing insights into their underlying causes and functional roles, which are crucial for unraveling their systemic impact.

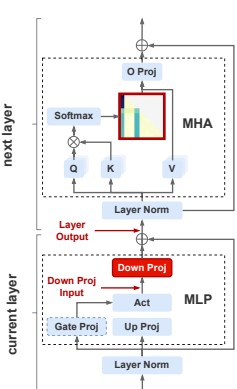

**Figure 2:** Illustration of systematic outliers locations in LLMs.

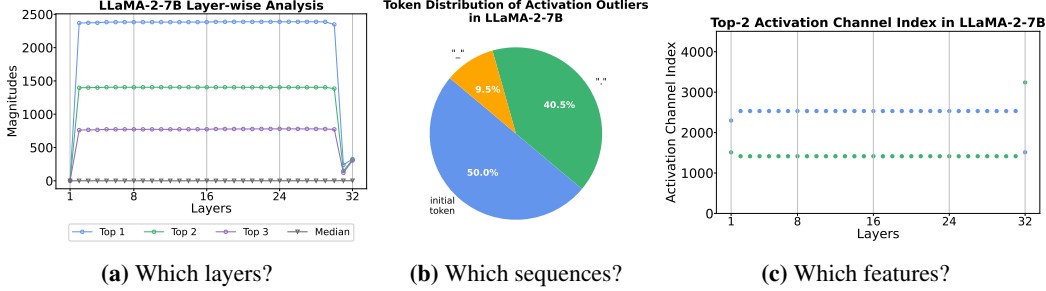

**(a)** Which layers? **(b)** Which sequences? **(c)** Which features?

**Figure 3:** Distribution of activation outliers in $\mathbf{h}_\ell$ across layers, sequences, and feature dimensions.

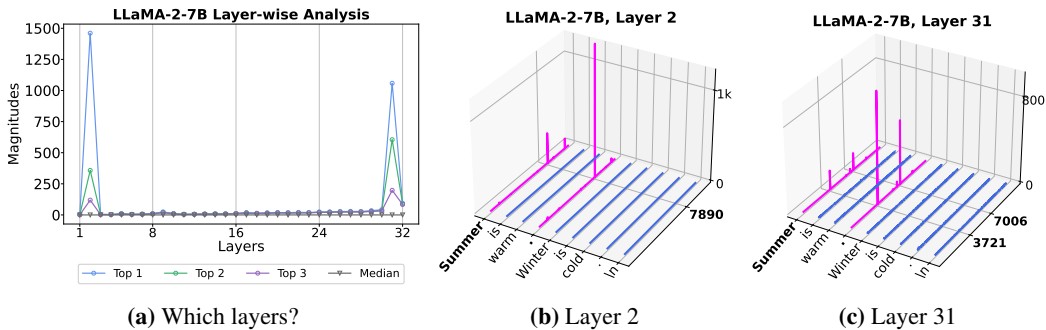

**(a)** Which layers? **(b)** Layer 2 **(c)** Layer 31

**Figure 4:** Distribution of activation outliers in $\mathbf{x}_\ell^{\text{down}}$ across layers, sequences, and feature dimensions.

## 3.1 WHERE ARE ACTIVATION OUTLIERS LOCATED?

Activation outliers manifest as abnormally large values in specific sequence and feature dimensions, as shown in Figures 1(a) and 1(c). These outliers appear in two distinct activation types: **layer outputs** $\mathbf{h}_\ell$ and **down-projection inputs** $\mathbf{x}_\ell^{\text{down}}$. We analyze their distributions across layers, sequences, and feature dimensions to understand their patterns. Detailed experimental settings are provided in Appendix B.1.

For layer outputs, Figure 3(a) reveals that activation outliers are concentrated in shallow layers, persist through middle layers, and diminish in the final layers. Additionally, Figure 3(b) shows that these outliers are associated with start tokens and weak semantic tokens, such as "." and "_", while Figure 3(c) highlights their confinement to specific feature dimensions.

For down-projection inputs, Figure 4 indicates that activation outliers are confined to a few shallow and deep layers. Similar to layer outputs, these outliers are linked to start tokens and weak semantic tokens and are concentrated in a limited set of feature dimensions.

In summary, activation outliers in $\mathbf{x}_\ell^{\text{down}}$ are restricted to shallow and deep layers, while those in $\mathbf{h}_\ell$ persist from shallow to middle layers. Both types predominantly affect fixed feature dimensions and tokens with weak semantic content.

## 3.2 WHERE ARE WEIGHT OUTLIERS LOCATED?

As shown in Figure 1(b), weight outliers are characterized by extreme values concentrated in specific columns. To quantify this, we compute the extremal ratio, defined as the ratio of the maximum value to the mean value within each column, since large column values directly influence the corresponding output activations.

Figure 5(a) illustrates the extremal ratio across layers and modules in LLaMA2-7B, highlighting that weight outliers are concentrated in the down-projection matrices $\mathbf{W}_\ell^{\text{down}}$ of the second layer and the last two layers. Figures 5(b) and 5(c) provide detailed visualizations of these outliers in the last two layers.

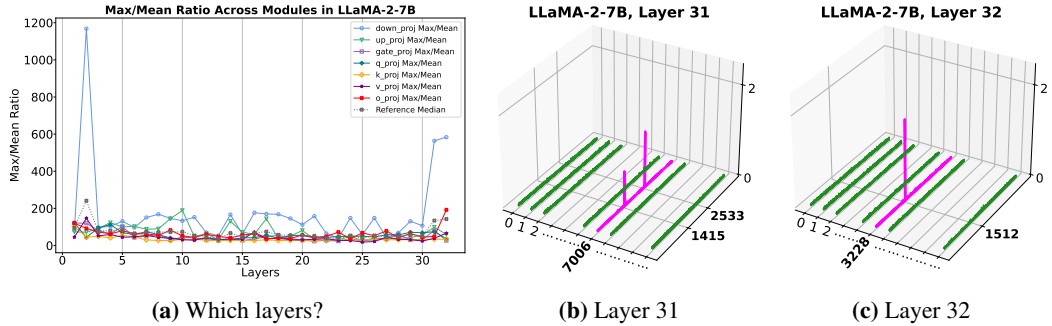

**Figure 5:** Distribution of weight outliers in $\mathbf{W}_\ell^{\text{down}}$ across layers, modules, and feature dimensions.

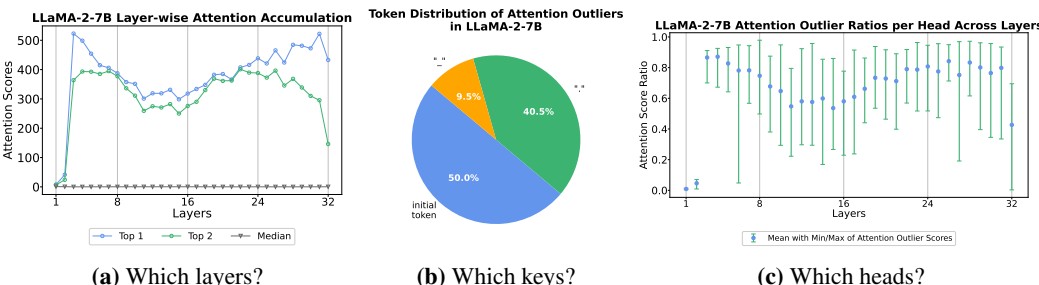

**Figure 6:** Distribution of attention outliers in $\mathbf{A}_\ell^i$ across layers, keys and heads.

In summary, weight outliers in LLaMA2-7B are primarily located in the MLP's down-projection matrices, concentrated in specific shallow and deep layers.

### 3.3 WHERE ARE ATTENTION OUTLIERS LOCATED?

To understand the distribution of attention outliers, we analyze cumulative attention scores on the attention weights $\mathbf{A}_\ell^i$ across layers, keys, and heads.

Figure 6 highlights key patterns in the distribution of attention outliers. In Figure 6(a), the two largest cumulative attention scores per layer show that attention outliers persist across all layers. Figure 6(b) categorizes keys with outliers, indicating a strong association with start tokens and weak semantic tokens. Figure 6(c) visualizes score fluctuations across heads, revealing significant variation between heads and layers.

## 4 SYSTEMATIC OUTLIERS ARE SIMULTANEOUS AND INTERCONNECTED

Systematic outliers are not isolated phenomena; instead, they exhibit strong correlations across feature and sequence dimensions as well as layers. Understanding these interconnections and their lifecycle is crucial for uncovering how outliers propagate through the model and affect computations. This section analyzes these relationships in detail, laying the groundwork for the next section, where we hypothesize and validate their functional roles.

### 4.1 HOW ARE THESE OUTLIERS RELATED?

**Table 1:** Consistency of different types of outliers across dimensions.

| Outlier Type 1 | Outlier Type 2 | Dimension | Consistency |
|---|---|---|---|
| weight outliers in $\mathbf{W}_\ell^{\text{down}}$ | activation outliers in $\mathbf{x}_\ell^{\text{down}}$ | feature | 100% |
| weight outliers in $\mathbf{W}_\ell^{\text{down}}$ | activation outliers in $\mathbf{h}_\ell$ | feature | 100% |
| activation outliers in $\mathbf{x}_\ell^{\text{down}}$ | activation outliers in $\mathbf{h}_\ell$ | sequence | 100% |
| activation outliers in $\mathbf{h}_\ell$ | attention outliers in $\mathbf{A}_\ell^i$ | sequence | 95% |

Table 1 provides a quantitative analysis of the alignment among the three types of outliers. Detailed experimental settings can be found in Appendix B.2. We find that the three types of outliers

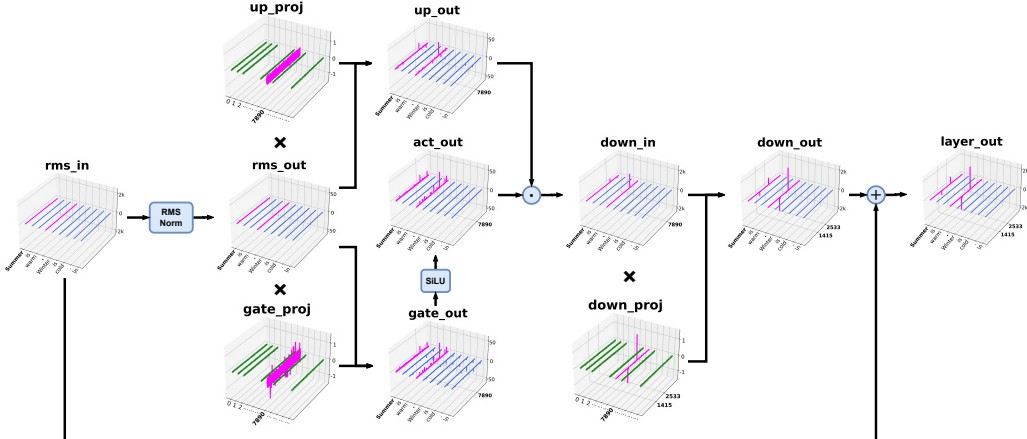

**Figure 7:** The emergence of activation outliers from weight outliers.

are not isolated but occur simultaneously and are interconnected across multiple dimensions within the model. Specifically, weight outliers align perfectly with activation outliers in feature dimensions, while over 95% of activation outliers overlap with attention outliers in sequence dimensions, primarily concentrated in start and weak semantic tokens.

### 4.2 THE LIFECYCLE OF SYSTEMATIC OUTLIERS

The correlations between different types of outliers suggest a deeper connection underlying their occurrences. By visualizing their lifecycle, we observe a chain of interactions: weight outliers lead to activation outliers, which then influence attention outliers, with this influence extending to non-outlier tokens.

**The Emergence of Activation Outliers from Weight Outliers.** In the second layer of LLaMA2-7B's MLP, weight outliers in the up- and gate-projection matrices cause extreme neuron responses. These are amplified by the SiLU activation function (Elfwing et al., 2018) and GLU operation (Shazeer, 2020), resulting in activation outliers that are up to a thousand times the average magnitude. Additionally, weight outliers in the down-projection matrix amplify activations in specific feature dimensions (e.g., 1415th and 2533rd), dominating the residual connection and influencing the final output (see Figure 7).

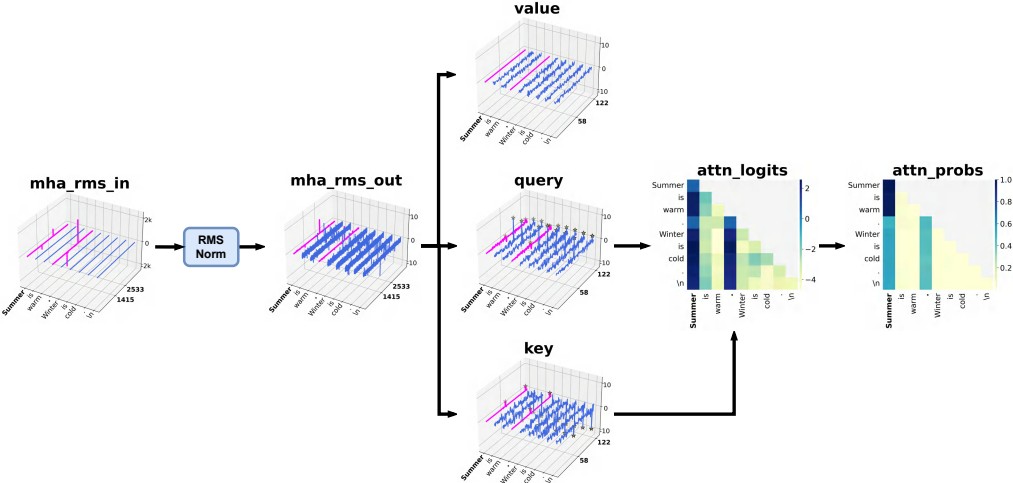

**Figure 8:** The spread of attention outliers from activation outliers. Activation outliers influence the self-attention mechanism, extending their impact to other sequence dimensions.

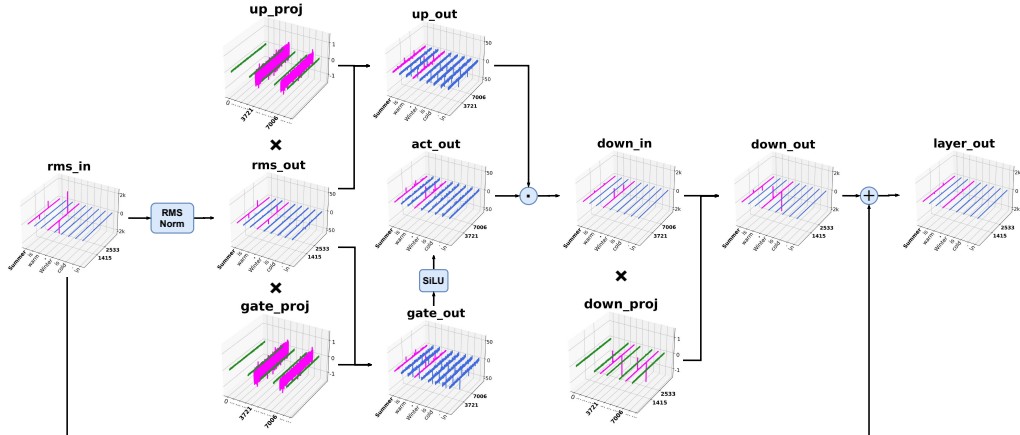

**Figure 9:** The disappearance of outliers in the final layers.

**The Spread of Attention Outliers from Activation Outliers.** Activation outliers propagate into the attention mechanism through their influence on query, key, and value vectors. In the third layer's Multi-Head Attention (MHA), we observe that tokens with activation outliers exhibit alignment in the 58th and 122nd dimensions of both **query** and **key vectors**. This alignment significantly increases the dot product between these vectors, leading to disproportionately high attention weights assigned to the outlier tokens (see Figure 8). As a result, the attention mechanism focuses heavily on these tokens, amplifying their influence across layers.

Interestingly, despite receiving significant attention, the **value vectors** corresponding to these tokens show comparatively smaller magnitudes. This indicates that while outlier tokens attract attention, they may contribute less directly to the final output. This phenomenon could imply a mechanism where the model leverages these tokens as "anchors" to affect attention outputs, we explore it further in Section 5.

Additionally, we find that this pattern—alignment in query and key dimensions and reduced magnitude in value dimensions—is consistent across most heads and layers. This consistency highlights the systematic nature of how activation outliers propagate their influence through the attention mechanism.

**The Disappearance of Outliers in the Final Layers.** Outliers gradually vanish in the final layers due to cancellation by values of opposite signs. This neutralization occurs progressively rather than abruptly (see Figure 9). As activation outliers diminish, attention outliers are similarly reduced, with some heads in the final layer showing no outliers at all.

**Summary.** In the lifecycle of systematic outliers, weight outliers drive the emergence of activation outliers, which propagate anomalies into the attention mechanism. This interdependence extends their influence to non-outlier tokens. These findings reveal that systematic outliers are intrinsically linked to the attention mechanism, setting the stage for the next section, where we hypothesize and validate their functional roles.

## 5 SYSTEMATIC OUTLIERS AS CONTEXT-AWARE SCALING FACTORS IN ATTENTION MECHANISMS

### 5.1 HYPOTHESES ON THE ROLE OF SYSTEMATIC OUTLIERS

Based on the observations in Section 4, we propose three hypotheses on the potential role of systematic outliers in LLMs, drawing from prior research and our findings:

1. **Fixed but Important Biases:** Inspired by the concept of Massive Activations (Sun et al., 2024), systematic outliers may act as fixed biases that consistently influence model behavior. These outliers could serve as stable values that help the model emphasize certain tokens or features, regardless of the context.

2. **Context-Aware Biases:** As seen in Figure 6(c), the attention outliers vary significantly across heads and tokens (20% to 95%). This suggests that these outliers may dynamically adjust their influence based on the input sequence, acting as context-aware signals that adapt to specific content and guide attention allocation.

3. **Context-Aware Scaling Factors:** Figure 8 shows that the value vectors corresponding to outlier tokens have significantly smaller magnitudes, suggesting that these outliers may act as implicit scaling factors. By reducing the impact of contextual information on certain tokens, these scaling factors help minimize unnecessary updates.

## 5.2 Empirical Validation of Systematic Outliers Hypotheses

**Formulation.** In this part, we introduce five different attention formulations to explore the role of systematic outliers. Each formulation represents a specific variant of the attention mechanism, designed to isolate different aspects of bias and scaling effects within the model. The formulations are listed in Table 2, followed by an explanation of their roles.

**Table 2:** Variants of the attention mechanism for systematic outliers analysis.

| ID | Attention Variant | Formulation |
|---|---|---|
| (a) | Default Attention (Vaswani, 2017) | $\mathrm{Attn}(Q, K, V) = \mathrm{softmax}\left(\frac{QK^T}{\sqrt{d}}\right) V$ |
| (b) | Explicit Fixed Bias | $\mathrm{Attn}(Q, K, V; \mathbf{v}') = \mathrm{softmax}\left(\frac{QK^T}{\sqrt{d}}\right) V + \mathbf{v}'$ |
| (c) | Explicit Context-Aware Bias | $\mathrm{Attn}(Q, K, V; \mathbf{k}', \mathbf{v}') = \mathrm{softmax}\left(\frac{QK^T}{\sqrt{d}}\right) V + \mathrm{softmax}\left(\frac{Q[K^T\ \mathbf{k}']}{\sqrt{d}}\right)\begin{bmatrix} 0^T \\ \mathbf{v}'^T \end{bmatrix}$ |
| (d) | Attention Bias | $\mathrm{Attn}(Q, K, V; \mathbf{k}', \mathbf{v}') = \mathrm{softmax}\left(\frac{Q[K^T\ \mathbf{k}']}{\sqrt{d}}\right)\begin{bmatrix} V \\ \mathbf{v}'^T \end{bmatrix}$ |
| (e) | Explicit Context-Aware Scaling Factor | $\mathrm{Attn}(Q, K, V) = S_c(x) \cdot \mathrm{softmax}\left(\frac{QK^T}{\sqrt{d}}\right) V$ |

In Table 2, $Q, K, V \in \mathbb{R}^{T \times d}$ are the query, key, and value matrices, and $d$ is the dimensionality of the hidden space. For variants with bias or scaling modifications, $\mathbf{k}', \mathbf{v}' \in \mathbb{R}^d$ represent the additional learnable bias terms, and $S_c(x)$ is a learnable scaling factor dependent on the input context.

- **Default Attention (a):** The standard attention mechanism serves as the baseline, without bias or scaling modifications.

- **Explicit Fixed Bias (b):** Adds a fixed, learnable bias $\mathbf{v}'$ only to the value matrix to isolate the impact of fixed biases on systematic outliers.

- **Explicit Context-Aware Bias (c):** Introduces context-aware bias terms $\mathbf{k}'$ and $\mathbf{v}'$, which vary based on the input sequence to isolate the impact of context-aware bias on systematic outliers.

- **Attention Bias (d):** Incorporates learnable bias terms $\mathbf{k}'$ and $\mathbf{v}'$ into the key and value matrices. It provides both context-aware bias and a scaling factor.

- **Explicit Context-Aware Scaling Factor (e):** Utilizes a learnable scaling factor $S_c(x)$ that dynamically adjusts the attention weights, helping investigate the scaling effect on reducing systematic outliers.

**Results.** We train five GPT-2 (Radford et al., 2019) models with these attention variants. Detailed experimental settings can be found in Appendix B.3. We visualize the presence of activation outliers across different attention formulations in Figure 10 and plotted the top-3 largest activation magnitudes for each layer in Figure 11. The results reveal distinct patterns, where only attention bias (d) and explicit context-aware scaling factor (e) effectively prevent the formation of systematic outliers.

These results strongly suggest that scaling factor plays a crucial role in managing outlier behavior in LLMs. Fixed or context-aware bias alone is insufficient to mitigate outliers, whereas explicit context-aware scaling factor provides the necessary dynamic adjustment to prevent their occurrence.

The main conclusions about the role of systematic outliers from the experiments are as follows:

- **It is not the fixed bias:** Experiment (b) clearly shows that fixed bias alone does not prevent outliers, proving that fixed bias mechanisms are ineffective in this regard.

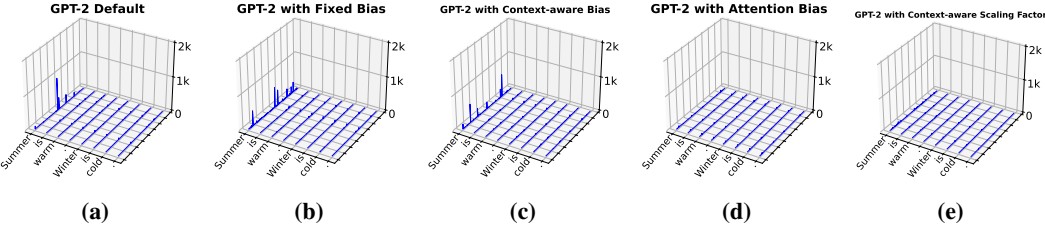

**Figure 10:** Activation outliers across different attention formulations.

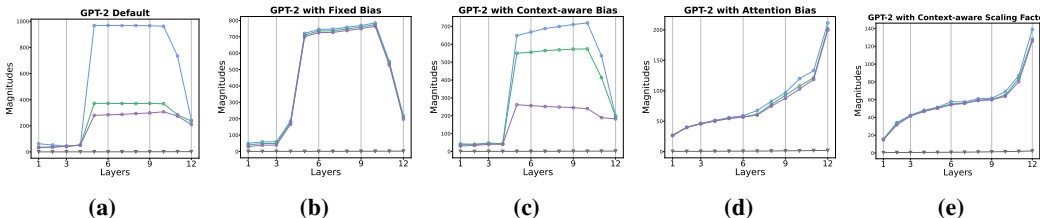

**Figure 11:** Top-3 largest activation outliers for each layer.

- **It is not the context-aware bias:** Experiment (c) shows that context-aware bias alone is insufficient. A comparison with Experiment (d) highlights that context-aware biases do not eliminate outliers.

- **It is the context-aware scaling factor:** Both Experiment (d) and Experiment (e) explicitly use a context-aware scaling factor. While Experiment (d) includes additional context-aware bias, the comparison between (c) and (d) confirms that the primary factor in preventing systematic outliers is the context-aware scaling factor.

## 5.3 FURTHER ANALYSIS OF SYSTEMATIC OUTLIERS

**Softmax Attention is the root cause of systematic outliers.** In Transformers, multi-head attention (MHA) aims to augment token embeddings with contextual information to improve prediction accuracy. The difficulty of this task varies across tokens: some require complex contextual updates (e.g., for ambiguous or semantically rich tokens), while others (e.g., delimiter or padding tokens) require minimal updates. However, the softmax operation enforces that attention scores always sum to one, even for simpler tasks where little contextual information is needed. To satisfy this constraint, the model must produce a large dynamic range in the input to softmax, amplifying the disparity between tokens. This dynamic range is further exaggerated as the model learns to allocate most attention to low-information tokens in some cases, ensuring minimal updates for those tokens.

This demand for a large dynamic range propagates through the network and affects earlier layers. Layer Normalization, applied before softmax, standardizes input distributions, inadvertently compressing the required dynamic range. To compensate, multi-layer perceptron (MLPs) in preceding layers generate activations of significantly higher magnitude, resulting in the emergence of activation outliers. These high-magnitude activations propagate through the residual connections, amplifying gradients during backpropagation and encouraging the formation of weight outliers in the projection matrices.

Furthermore, because softmax outputs are strictly positive and sum to one, all tokens maintain non-zero probabilities, leading to persistent gradients for all queries and keys. This forces the model to continuously adjust activations and weights to accommodate large dynamic ranges, further amplifying systematic outliers over successive layers. As a result, softmax normalization, coupled with architectural constraints like Layer Normalization and residual connections, becomes the fundamental driver of systematic outliers. Detailed derivations and analysis are provided in Appendix C.

**Potential Applications for Model Compression.** Systematic outliers complicate compression techniques such as quantization and pruning by increasing memory usage and degrading performance. Our experiments demonstrate that context-aware scaling factors effectively mitigate these

issues. Table 3 compares GPT-2 Default and GPT-2 with Context-aware Scaling Factor under common compression methods.

**Table 3:** Comparison of GPT-2 Default and GPT-2 with Context-aware Scaling Factor under pruning and quantization methods.

| Model | PPL (FP16) | PPL (AbsMax W8) | PPL (50% Sparse) |
|---|---|---|---|
| GPT-2 Default | 27.24 | 93.44 | 7235.68 |
| GPT-2 + Context-aware Scaling | 26.95 | 29.22 | 39.47 |

The results, tested on WikiText2, show that context-aware scaling significantly improves robustness to compression. For quantization, it reduces PPL from 93.44 to 29.22, and for pruning, it stabilizes PPL at 39.47 compared to 7235.68 for GPT-2 Default. Importantly, the addition of context-aware scaling factors incurs minimal memory overhead. For example, in GPT-2, the parameter count increases from 123.59M to 123.70M—less than a 0.1% increase. This negligible overhead ensures that the method remains practical for large-scale deployment. These findings validate that context-aware scaling enhances model robustness to compression, enabling efficient model deployment without compromising performance.

**Influence on Convergence and Training Stability.** Incorporating context-aware scaling factors significantly accelerates convergence and enhances training stability in Transformer models. By dynamically adjusting attention scores, these factors reduce reliance on extreme outliers, leading to a smoother optimization landscape. As shown in Figure 12, models with context-aware scaling converge faster during early training steps, but their final validation losses are comparable to those of default attention mechanisms, indicating improved convergence speed without necessarily better final performance.

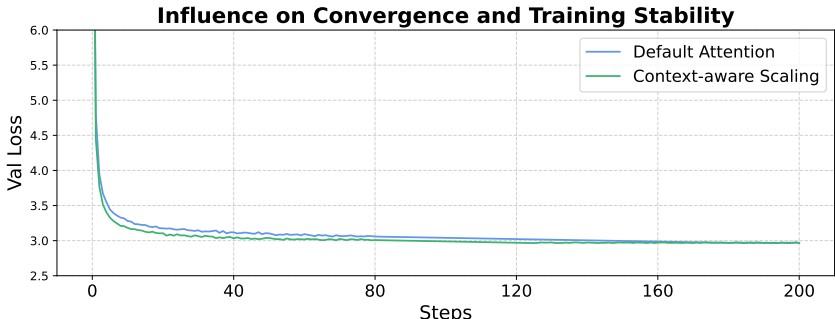

**Figure 12:** Training loss comparison showing improved convergence by eliminating outliers.

## 6 CONCLUSION

In this work, we systematically analyzed the distribution, formation, and roles of outliers in large language models (LLMs), categorizing them into activation, weight, and attention outliers. Our findings reveal that these outliers are interconnected across layers and dimensions, stemming from the softmax operation in the self-attention mechanism. Acting as implicit, context-aware scaling factors, these outliers dynamically adjust attention distributions, enabling the model to balance diverse contextual demands. By eliminating these outliers through explicit context-aware scaling, we showed improvements in model convergence and compression efficiency. Our approach helps reduce unnecessary attention allocation, making models more efficient and stable. This finding provides new insights into the internal workings of Transformer-based LLMs and opens up avenues for refining attention mechanisms to improve performance and efficiency. We believe that our study not only deepens the theoretical understanding of outliers in LLMs but also has practical implications for the development of more efficient and robust language models.

## 7 ACKNOWLEDGEMENTS

This work was supported by National Key R&D Program of China under Grant No.2023ZD0120400, Beijing Municipal Science and Technology Project (Z231100007423004),

Zhejiang Lab (No. 2021KH0AB07), and National Natural Science Foundation of China (Grant No. 62206290, 62276260, 62176254, 61976210, 62076235).

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

# A    ADDITIONAL RESULTS ON SYSTEMATIC OUTLIERS IN LLMS

This section extends the analysis of systematic outliers in LLMs, complementing the findings presented in the main paper. We provide additional results on both pretrained LLMs (Appendix A.1) and fine-tuned variants (Appendix A.2), highlighting the consistency and variation in outlier behavior across different model families and training paradigms.

## A.1    PRETRAINED LLMS

In Section 3, we identified systematic outliers in models such as LLaMA2-7B. To further validate these findings, we extend our evaluation to a broader set of pretrained LLMs, including Phi-2 (Javaheripi & Bubeck, 2023), Mistral-7B (Jiang et al., 2023), LLaMA2-13B, LLaMA3 (Dubey et al., 2024), OPT-6.7B (Zhang et al., 2022), MPT-7B (MosaicML, 2023), and Falcon-7B (Almazrouei et al., 2023). The corresponding results are depicted in Figure 15, 16, 17, 18 and 19.

Our analysis reveals several key insights:

- **Consistency of Outliers:** Systematic outliers consistently appear across all evaluated models, with patterns similar to those observed in Section 3.

- **Influence of Model Architecture:** The design of the MLP architecture and the choice of activation function significantly impact the distribution of weight outliers and activation outliers. For example, OPT-6.7B, which employs a two-layer linear structure with ReLU activation, exhibits a more dispersed pattern of outliers compared to the compact clustering seen in models like LLaMA2-13B and Mistral-7B.

- **Layer-Specific Trends:** In deeper layers, particularly in LLaMA3 and Falcon-7B, outliers in down-projection activations are more pronounced, suggesting that architectural modifications in newer models can amplify outlier formation in specific layers.

These results demonstrate that systematic outliers are a pervasive phenomenon across diverse model families and architectures.

## A.2    FINE-TUNED LLMS

Beyond pretrained models, fine-tuning plays a crucial role in adapting LLMs for specific tasks such as instruction following or conversational applications (Ouyang et al., 2022). To assess the impact of fine-tuning on systematic outliers, we analyze fine-tuned variants of LLaMA2 and Mistral, with results shown in Figures 20, 21, and 22.

- **Persistence of Outliers:** Systematic outliers persist after fine-tuning, with their magnitudes and distributions largely unchanged from the corresponding pretrained models. For instance, Figures 14 and 22 demonstrate that outlier patterns in Mistral-7B are consistent with its fine-tuned counterpart, Mistral-7B Instruct. Similarly, Figures **??** and 20 show that fine-tuning LLaMA2-7B into LLaMA2-7B-Chat introduces minimal changes to outlier locations and distributions.

- **Fine-Tuning Effects:** Instruction fine-tuning does not significantly alter the structural patterns of systematic outliers. Although minor variations in magnitudes or attention scores are observed in some layers, such as reduced attention outlier intensities in Mistral-7B Instruct, these adjustments do not affect the overall spatial or dimensional concentration of outliers.

- **Generalizability:** The consistent presence of systematic outliers across both pretrained and fine-tuned models indicates a structural origin rooted in the Transformer architecture, rather than task-specific artifacts introduced during fine-tuning. This highlights their intrinsic nature and robustness to different training paradigms.

These results highlight the robustness of systematic outliers across both pretrained and fine-tuned models, emphasizing their structural roots within Transformer architectures.

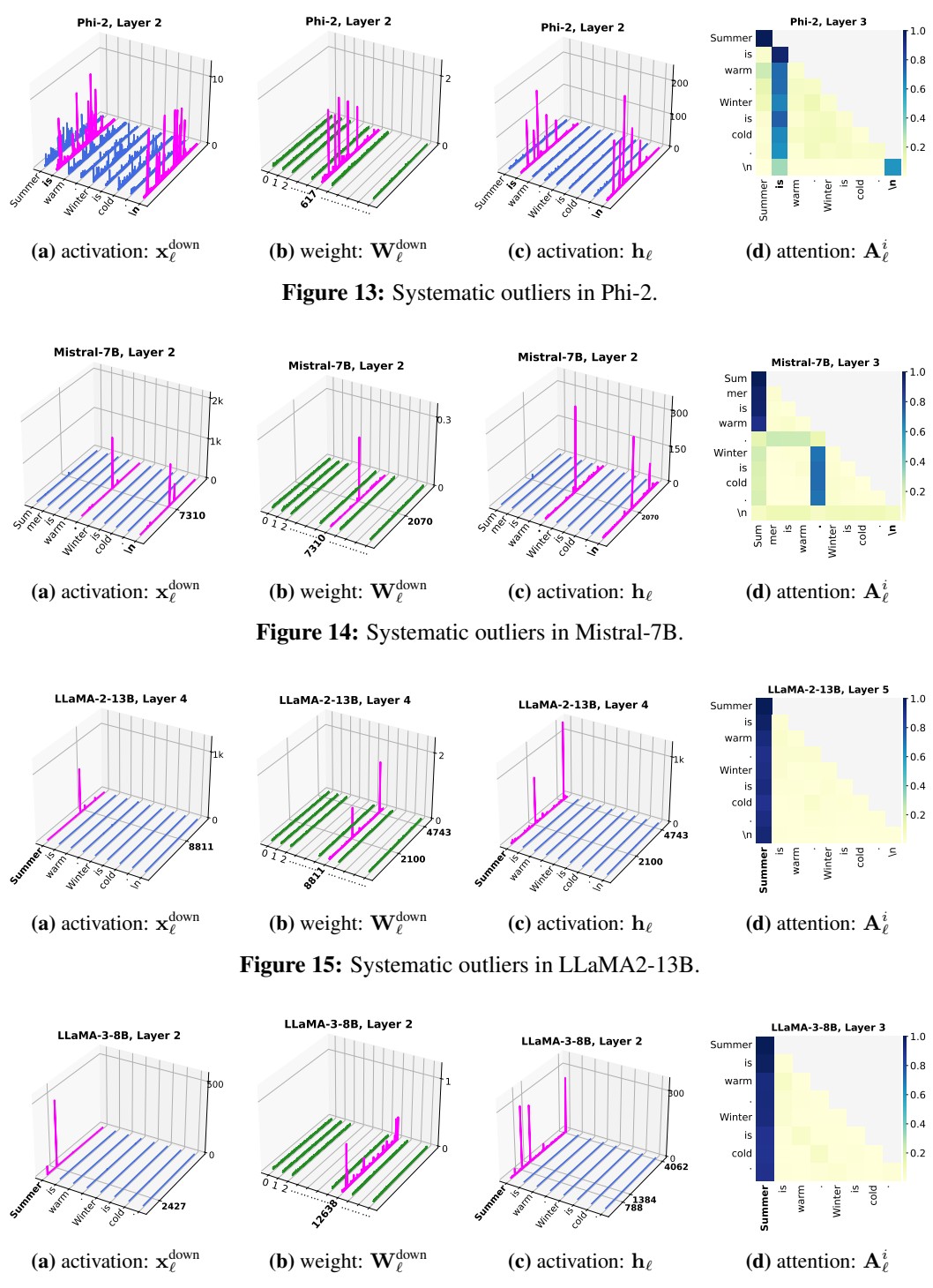

**Figure 13:** Systematic outliers in Phi-2.

**Figure 14:** Systematic outliers in Mistral-7B.

**Figure 15:** Systematic outliers in LLaMA2-13B.

**Figure 16:** Systematic outliers in LLaMA3-8B.

(a) activation: $\mathbf{x}_\ell^{\text{down}}$     (b) weight: $\mathbf{W}_\ell^{\text{down}}$     (c) activation: $\mathbf{h}_\ell$     (d) attention: $\mathbf{A}_\ell^i$

## B  DETAILED EXPERIMENTAL SETTINGS

This section provides comprehensive details on the experimental settings used to analyze systematic outliers in LLMs. It covers the methodologies and configurations employed in different aspects of the study, ensuring reproducibility and clarity. The following subsections provide detailed descriptions of the experimental setups for each analysis.

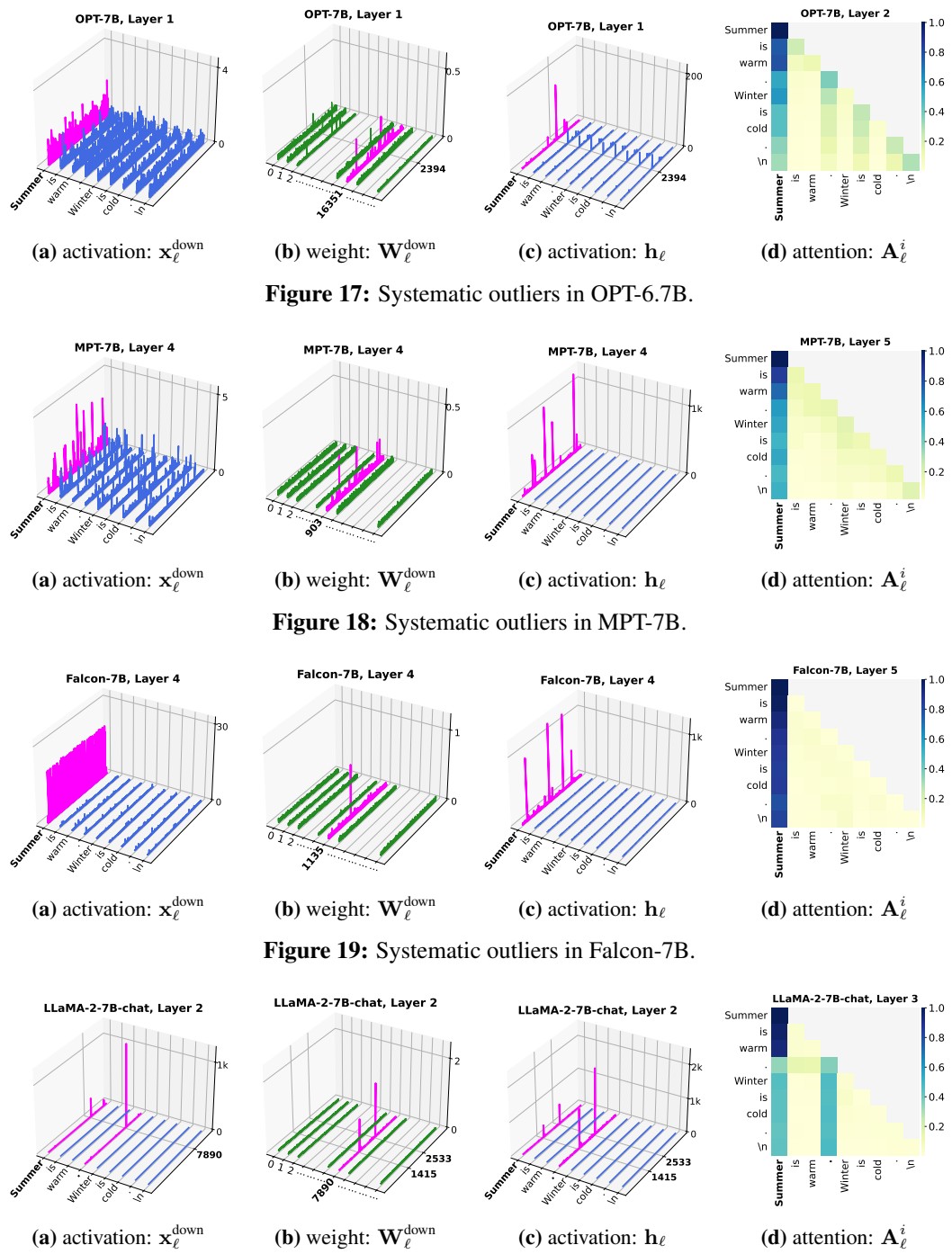

**Figure 17:** Systematic outliers in OPT-6.7B.

(a) activation: $\mathbf{x}_\ell^{\text{down}}$  (b) weight: $\mathbf{W}_\ell^{\text{down}}$  (c) activation: $\mathbf{h}_\ell$  (d) attention: $\mathbf{A}_\ell^i$

**Figure 18:** Systematic outliers in MPT-7B.

(a) activation: $\mathbf{x}_\ell^{\text{down}}$  (b) weight: $\mathbf{W}_\ell^{\text{down}}$  (c) activation: $\mathbf{h}_\ell$  (d) attention: $\mathbf{A}_\ell^i$

**Figure 19:** Systematic outliers in Falcon-7B.

(a) activation: $\mathbf{x}_\ell^{\text{down}}$  (b) weight: $\mathbf{W}_\ell^{\text{down}}$  (c) activation: $\mathbf{h}_\ell$  (d) attention: $\mathbf{A}_\ell^i$

**Figure 20:** Systematic outliers in LLaMA2-7B-Chat.

## B.1 POSITION ANALYSIS SETTINGS

This subsection describes the experimental settings used to analyze the distribution of systematic outliers in LLaMA2-7B across layers, sequences, and feature dimensions. The focus is on three types of outliers—**activation outliers**, **weight outliers**, and **attention outliers**, with results presented in Figures 3, 4, 5, and 6.

For activation outliers in $\mathbf{h}_\ell$ (layer outputs) and $\mathbf{x}_\ell^{\text{down}}$ (down-projection inputs), we analyze the top-3 largest activation values and the median activation value for each layer to examine how outliers

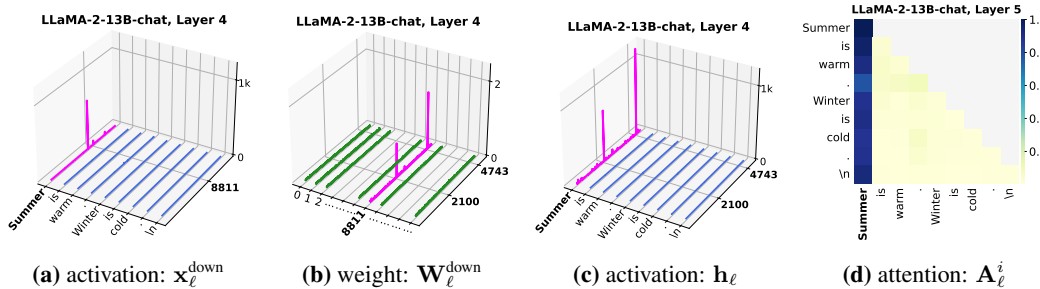

**(a)** activation: $\mathbf{x}_\ell^{\text{down}}$    **(b)** weight: $\mathbf{W}_\ell^{\text{down}}$    **(c)** activation: $\mathbf{h}_\ell$    **(d)** attention: $\mathbf{A}_\ell^i$

**Figure 21:** Systematic outliers in LLaMA2-13B-Chat.

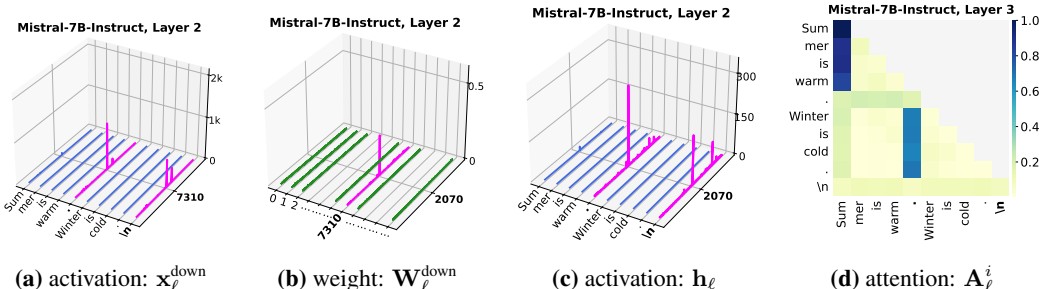

**(a)** activation: $\mathbf{x}_\ell^{\text{down}}$    **(b)** weight: $\mathbf{W}_\ell^{\text{down}}$    **(c)** activation: $\mathbf{h}_\ell$    **(d)** attention: $\mathbf{A}_\ell^i$

**Figure 22:** Systematic outliers in Mistral-7B-Instruct.

are distributed across shallow, middle, and deep layers. Using 100 sequences of length 2,048 from the RedPajama dataset (Computer, 2023), the positions where activation outliers first appear are identified. Additionally, scatter plots visualize the channel indices where activation outliers occur, highlighting fixed feature dimensions associated with these outliers.

For weight outliers in $\mathbf{W}_\ell^{\text{down}}$ (down-projection matrices), we compute the *extremal ratio* for each layer and module to measure how concentrated the largest weights are in specific columns.

For attention outliers in $\mathbf{A}_\ell^i$ (attention weights), we analyze the top-2 cumulative attention scores and the median score for each layer to track how attention outliers persist across layers. Sequence positions where attention outliers first appear are identified using the same set of 100 sequences. Finally, the mean, upper limit, and lower limit of attention outlier scores are computed for different heads across layers to reveal head-specific patterns.

These settings provide a systematic approach to understanding the spatial and dimensional patterns of systematic outliers in LLaMA2-7B.

## B.2 Consistency Analysis Settings

To analyze the consistency between different types of outliers, we calculate the alignment of activation and attention outliers across 100 randomly selected samples from the RedPajama dataset (Computer, 2023). Each sample has a sequence length of 2,048, and attention outliers are analyzed separately for each attention head.

The overlap between activation and attention outliers is defined as the percentage of sequence indices where activation outliers in $\mathbf{h}_\ell \in \mathbb{R}^{\text{seqlen} \times d_{\text{hidden}}}$ align with attention outliers in $\mathbf{A}_\ell^i \in \mathbb{R}^{\text{seqlen} \times \text{seqlen}}$, where $\mathbf{A}_\ell^i$ is the cumulative attention score matrix for head $i$ at layer $\ell$. For each sample and attention head, overlaps are computed as:

$$\text{Overlap}^i = \frac{|\mathcal{O}_{\text{activation}} \cap \mathcal{O}_{\text{attention}}^i|}{|\mathcal{O}_{\text{activation}}|},$$

where $\mathcal{O}_{\text{activation}}$ and $\mathcal{O}_{\text{attention}}^i$ are the sets of sequence indices corresponding to activation and attention outliers, respectively. The overall overlap across $n_{\text{samples}}$ samples and $n_{\text{heads}}$ attention heads is then averaged as:

$$\text{Overall Overlap} = \frac{1}{n_{\text{samples}} \cdot n_{\text{heads}}} \sum_{k=1}^{n_{\text{samples}}} \sum_{i=1}^{n_{\text{heads}}} \text{Overlap}^i.$$

These settings systematically quantify the alignment between activation and attention outliers, offering detailed insights into their interconnected nature, as summarized in Table 1.

### B.3 GPT-2 Attention Variant Training Settings

We utilize the open-source GPT-2 implementation from the NanoGPT repository (Karpathy, 2023), following the default recommended training setup and optimizer settings. Each of the five GPT-2 variants was trained for 50,000 iterations, processing approximately 2 billion tokens in total. For the attention bias variant, we followed the initialization method proposed by (Sun et al., 2024), setting $k'$ and $v'$ to $\mathcal{N}(0, 0.02\mathbf{I})$.

### B.4 Model Compression Experiment Settings

To evaluate the robustness of context-aware scaling factors under compression, we conducted experiments on GPT-2 models using two common methods: quantization and pruning. For quantization, 8-bit weight quantization was applied using the AbsMax scaling method, which normalizes weights by their maximum absolute value. For pruning, we performed unstructured magnitude pruning, removing 50% of the smallest-magnitude weights across all layers.

Both models were evaluated on the WikiText2 dataset using perplexity (PPL) as the primary metric, comparing GPT-2 Default and GPT-2 with context-aware scaling factors. These settings were chosen to test the models' ability to maintain performance under aggressive compression techniques.

## C How Softmax Causes Systematic Outliers in Transformer Models

The formation of systematic outliers in transformer models stems from the inherent characteristics of the softmax operation within the self-attention mechanism. While capturing the complete dynamics of outlier emergence requires a complex understanding of training processes, we provide a mathematical analysis that outlines the logical connection between softmax and the appearance of systematic outliers. This analysis reveals how the interaction between softmax and model architecture propagates and localizes these anomalies. The following sequence summarizes the key steps:

1. **Necessity of Zero-Update in MHA:** Certain tokens, such as initial tokens or weakly semantic tokens, often require minimal contextual updates. The Multi-Head Attention (MHA) mechanism addresses this by dynamically suppressing updates for these tokens, placing strict constraints on gradients and weights.

2. **Softmax-Induced Dynamic Range Expansion:** Achieving the zero-update behavior requires softmax to focus attention weights on a limited number of keys. This necessitates substantial differences in the dot products of query and key vectors, leading to extreme dynamic ranges in the attention scores.

3. **Propagation of Systematic Outliers:** The extreme attention scores propagate anomalies through transformer computations:
   - In MHA, the shared projection weights for keys ($W_K$) and values ($W_V$) accumulate steep gradients, introducing activation anomalies.
   - In the Multi-Layer Perceptron (MLP), these activation anomalies are amplified by the projection layers, resulting in systematic outliers.

4. **Localization of Outliers:** Outliers concentrate at specific tokens and feature dimensions:

- **Token-Level:** Initial tokens (e.g., [CLS]) and weakly semantic tokens exhibit pronounced updates due to their unique roles in aggregation and suppression.
- **Channel-Level:** Outliers appear in a limited set of feature dimensions, reducing their impact on other parts of the model but exacerbating localized anomalies.

5. **Why Earlier Layers Avoid This:** Early transformer layers focus on distributing token information uniformly, maintaining balanced attention distributions. Systematic outliers predominantly emerge in deeper layers, where higher-level semantic differentiation sharpens the attention mechanism.

This systematic analysis reveals that the softmax operation, in combination with architectural constraints, is a central factor driving the emergence and propagation of systematic outliers. The subsequent sections delve into the specific mechanisms behind these phenomena, starting with the necessity of zero-update behavior in self-attention.

## C.1 NECESSITY OF ZERO-UPDATE IN MULTI-HEAD ATTENTION (MHA)

In transformer models, the Multi-Head Attention (MHA) mechanism dynamically adjusts token representations based on contextual information. However, certain tokens—such as initial tokens (e.g., [CLS]) or weakly semantic tokens (e.g., punctuation marks)—often require minimal updates during training. For these tokens, the desired behavior is a near-zero update:

$$\Delta x = \text{MHA}(Q_x, K, V) \approx 0.$$

Achieving this condition imposes constraints on the gradients and attention weights. The softmax normalization within MHA must allocate most of the attention probability to a few keys with negligible values, effectively canceling the contributions of the remaining keys. The output of the attention mechanism is given by:

$$\text{MHA}(Q, K, V) = \text{Softmax}\left(\frac{QK^\top}{\sqrt{d_k}}\right)V,$$

where $Q, K, V$ are the query, key, and value matrices, and $d_k$ is the dimensionality of the keys. For a given query $Q_x$, the attention weights must satisfy:

$$\sum_{j=1}^{n} \text{Softmax}\left(\frac{Q_x K_j^\top}{\sqrt{d_k}}\right) = 1.$$

To approximate zero-update, the weighted sum of $V_j$ values must effectively cancel out. This necessitates a highly concentrated attention weight distribution, which in turn creates large disparities in the query-key dot products ($Q_x K_j^\top$).

This zero-update requirement, while essential for certain tokens, introduces challenges:

- **Dynamic Weight Adjustment:** The softmax must focus the attention weights on specific keys, requiring the associated dot products ($Q_x K_j^\top$) to dominate.
- **Gradient Amplification:** The backpropagation process must enforce selective updates to queries ($Q_x$) and keys ($K_j$), leading to steep gradients.

This forms the foundation for the emergence of extreme values in both the attention weights and the gradients, as discussed in the following section.

## C.2 SOFTMAX-INDUCED DYNAMIC RANGE EXPANSION

The softmax operation, central to the self-attention mechanism, inherently expands the dynamic range of attention scores. This behavior is critical for satisfying the zero-update condition for certain tokens but simultaneously leads to the emergence of extreme values.

The softmax operation is defined as:

$$A_{ij} = \text{Softmax}\left(\frac{Q_i K_j^\top}{\sqrt{d_k}}\right) = \frac{\exp\left(\frac{Q_i K_j^\top}{\sqrt{d_k}}\right)}{\sum_{k=1}^n \exp\left(\frac{Q_i K_k^\top}{\sqrt{d_k}}\right)},$$

where $A_{ij}$ represents the attention weight assigned to key $j$ by query $i$, and $n$ is the sequence length. To achieve zero-update for a token $x$, the softmax must concentrate the attention weight $A_{xj}$ on specific keys $j^*$ while suppressing the weights for others. This requires the dot product $\frac{Q_x K_{j^*}^\top}{\sqrt{d_k}}$ to significantly dominate over other terms $\frac{Q_x K_k^\top}{\sqrt{d_k}}$. Mathematically, this results in:

$$\frac{Q_x K_{j^*}^\top}{\sqrt{d_k}} \gg \frac{Q_x K_k^\top}{\sqrt{d_k}}, \quad \forall k \neq j^*.$$

As a consequence, the ratio of exponential terms grows exponentially with the difference in dot products. For example, if $\frac{Q_x K_{j^*}^\top}{\sqrt{d_k}} = M$ and $\frac{Q_x K_k^\top}{\sqrt{d_k}} = 0$ for $k \neq j^*$, the resulting attention weight is:

$$A_{xj^*} = \frac{\exp(M)}{\exp(M) + (n-1)} \approx 1, \quad \text{as } M \to \infty.$$

In contrast, the weights for other keys approach zero:

$$A_{xk} = \frac{1}{\exp(M) + (n-1)} \quad \text{for } k \neq j^*.$$

This dynamic range expansion, driven by the softmax operation, imposes two key challenges:

- **Extreme Values in Attention Scores:** The attention weights for dominant keys grow exponentially, while others become negligible. This leads to highly imbalanced attention distributions.
- **Gradient Amplification:** Backpropagation through the softmax induces steep gradients for the dominant keys, amplifying updates to the query ($Q$) and key ($K$) vectors. This can destabilize the training process.

These extreme values propagate through subsequent layers, contributing to the formation of systematic outliers in both activations and weights, as detailed in the next subsection.

### C.3 PROPAGATION OF SYSTEMATIC OUTLIERS

Systematic outliers, once introduced by the softmax mechanism, propagate through the transformer layers due to shared weights and non-linear transformations in the Multi-Head Attention (MHA) and Multi-Layer Perceptron (MLP) components.

**Impact in MHA.** The shared projection weights in MHA—specifically, the key and value matrices $W_K$ and $W_V$—magnify the effect of extreme attention scores. The query ($Q$), key ($K$), and value ($V$) matrices are computed as:

$$Q = \text{LN}(h_\ell)W_Q, \quad K = \text{LN}(h_\ell)W_K, \quad V = \text{LN}(h_\ell)W_V,$$

where $h_\ell$ represents the layer's input, and LN is layer normalization. When the attention scores $A$ are dominated by a few keys, the output of MHA focuses heavily on the corresponding values $V_j$. The MHA output for token $x$ is given by:

$$\text{MHA}(x) = \sum_{j=1}^{n} A_{xj} V_j,$$

where $A_{xj}$ concentrates on a few dominant keys. This imbalance introduces large updates to the projection weights $W_K$ and $W_V$ during backpropagation, as the gradients for these weights are computed from $\nabla_{W_K} L$ and $\nabla_{W_V} L$, respectively.

**Amplification in MLP.** Following MHA, the output passes through the MLP block, which consists of up-projection ($W_{\text{up}}$), a non-linear activation ($\sigma$), and down-projection ($W_{\text{down}}$):

$$z_{\text{down}} = W_{\text{down}} \sigma(W_{\text{up}} \text{LN}(h_{\ell+1/2})),$$

where $h_{\ell+1/2} = \text{LN}(h_\ell) + \text{MHA}(\text{LN}(h_\ell))$. If $\text{MHA}(x)$ produces extreme values due to attention outliers, these anomalies are further amplified by the non-linear activation ($\sigma$) and concentrated in the down-projection weights ($W_{\text{down}}$).

**Emergence of Outliers.** The combination of steep gradients and non-linear transformations results in the emergence of outliers in both activations and weights. Key observations include:

- **Activation Outliers:** These appear in the intermediate representations $h_{\ell+1}$, primarily concentrated in specific sequence positions and feature dimensions.
- **Weight Outliers:** The steep gradients for $W_K$, $W_V$, and $W_{\text{down}}$ lead to concentrated large weights in these matrices, reinforcing the cycle of extreme values.

This propagation mechanism underscores the role of softmax-induced dynamic range expansion in perpetuating systematic outliers throughout the transformer layers.

### C.4 LOCALIZATION OF SYSTEMATIC OUTLIERS

Systematic outliers are not randomly distributed but exhibit specific patterns of localization across tokens, layers, and feature dimensions. This section explores how these outliers are concentrated at particular positions and channels, minimizing their overall disruption while fulfilling the model's dynamic range requirements.

**Token-Level Localization.** Outliers are predominantly associated with specific tokens, such as initial tokens (e.g., [CLS]) or tokens with weak semantic content (e.g., punctuation marks). These tokens are particularly susceptible to outliers due to their roles in aggregating sequence information or carrying minimal intrinsic meaning. For instance:

- **Initial Tokens:** Initial tokens aggregate global information, receiving disproportionately high attention scores, which amplifies their values in the MHA output.
- **Weak-Semantics Tokens:** Tokens like . or – often have low intrinsic information, leading the model to assign high attention scores to stabilize their contextual representation. This results in exaggerated updates during training.

**Channel-Level Localization.** Outliers in feature dimensions are typically confined to a small subset of channels. This sparsity arises because the model prioritizes containing the impact of extreme values to a few dimensions rather than spreading them across the entire representation. Key characteristics include:

- **Fixed Dimensions:** Outliers are observed in specific rows or columns of weight matrices (e.g., $W_{\text{down}}$) across layers, suggesting a structural origin.
- **Robustness Preservation:** By localizing extreme values to a few channels, the model ensures that most dimensions remain stable, preserving the robustness of the overall representation.

### C.5 CONCLUSION

The mathematical analysis presented in this section demonstrates how the softmax operation in the self-attention mechanism is a key driver of systematic outliers in transformer models. By fulfilling the zero-update requirement for certain tokens, softmax induces extreme disparities in attention scores, leading to steep gradients and the emergence of outliers. These anomalies propagate through transformer layers, being amplified by shared projection weights and non-linear activations in the MLP, ultimately manifesting as systematic outliers in both activations and weights.

Moreover, these outliers exhibit distinct localization patterns, being concentrated at specific tokens and feature dimensions. This localization minimizes their overall disruption to the model while fulfilling the dynamic range demands imposed by the softmax mechanism. Understanding these dynamics offers valuable insights into the structural origins of systematic outliers, paving the way for mitigation strategies such as explicit context-aware scaling factors to prevent their formation and improve model robustness.

## D MORE ANALYSIS FO SYSTEMATIC OUTLIERS

### D.1 ABSENCE OF OUTLIERS IN SIGMOID ATTENTION

In Ramapuram et al. (2024), sigmoid self-attention was proposed as an alternative to traditional softmax-based attention, formulated in Equation 1. Unlike softmax, sigmoid attention independently maps each attention score to a value between 0 and 1, introducing a fixed bias term $b$ to adjust the sigmoid function's activation.

$$
\mathbf{SigmoidAttn}\left(\boldsymbol{X}\right) = \sigma\left(\boldsymbol{Q}\boldsymbol{K}^T / \sqrt{d_{qk}}\right)\boldsymbol{V},
$$
$$
\text{with } \sigma : u \mapsto \mathrm{sigmoid}(u+b) := \left(1 + e^{-(u+b)}\right)^{-1} \tag{1}
$$

A key distinction of sigmoid attention is its ability to output near-zero values for certain tokens. While this can lead to vanishing gradients for some inputs, it also eliminates the extreme dynamic range caused by softmax normalization, thereby mitigating the formation of systematic outliers.

To evaluate its impact, we trained a GPT-2 model with sigmoid attention using the same experimental setup described in Appendix B.3. Figure 23 illustrates that sigmoid attention successfully eliminates systematic outliers. This finding reinforces our hypothesis that softmax normalization is a primary cause of outliers in self-attention mechanisms.

**GPT-2 with Sigmoid Attenion**

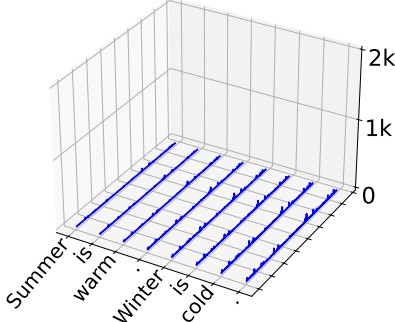

**Figure 23:** Absence of outliers in sigmoid attention.

### D.2 EXPLICIT CONTEXT-AWARE SCALING FACTOR IN TINY-LLAMA

To verify the generalizability of our findings beyond GPT-2, we conducted additional experiments using a TinyLLaMA-120M model. The training setup and code were adapted from the open-source implementation of TinyLLaMA (Keene, 2024), which provides an efficient framework for pretraining Transformer models.

In this experiment, we replaced the standard self-attention mechanism in TinyLLaMA with the "explicit context-aware scaling factor" variant. As shown in Figure 24, the results were consistent with those observed for GPT-2: systematic outliers were completely eliminated, confirming the effectiveness of the explicit context-aware scaling factor.

This finding further corroborates the analysis presented in the main paper. Standard attention mechanisms often require $\text{MHA}(x) \approx 0$ updates for certain tokens during training. Achieving this with softmax-based attention induces extremely large gradients, which lead to the formation of outliers in activations and weights. By contrast, the explicit context-aware scaling factor achieves the same zero-update objective without generating large gradients, thus avoiding outlier formation. These results demonstrate the robustness of the proposed approach across different Transformer architectures.

**TinyLLaMA with Context-aware Scaling Factor**

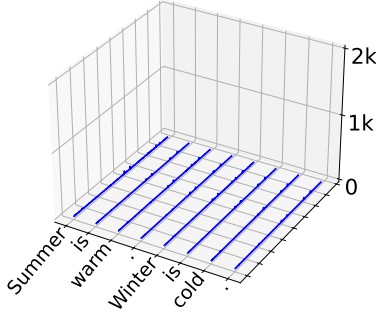

**Figure 24:** Visualization of TinyLLaMA-120M with explicit context-aware scaling factor. The absence of systematic outliers illustrates the effectiveness of this approach in preventing outlier formation.

### D.3 IMPACT OF SEQUENCE LENGTH ON OUTLIERS

Our analysis shows that sequence length does not affect the existence of attention outliers, but it does influence their specific positions within the sequence. Outliers consistently appear at the first tokens and semantically weak tokens, with their relative positioning shifting as the sequence length changes. The observations are summarized as follows:

- **Short Sequence:** In the sequence "Summer is warm!", attention outliers occur at "Summer".
- **Moderately Extended Sequence:** Extending the sequence to "Summer is warm! Winter is cold." introduces an additional outlier at "." (the first period), alongside "Summer".
- **Further Extended Sequence:** In the sequence "Summer is warm! Winter is cold. Spring is good.", outliers remain at "Summer" and the first period, demonstrating that outlier diversity does not increase with sequence length.
- **Modified Sequence:** Modifying the sequence to "Summer is warm! Winter is cold! Spring is good." shifts the outlier from the first period to the last period, while "Summer" and weak semantic tokens remain consistent outlier locations.

These examples suggest that while sequence length can shift the positions of outliers, their presence is robust across varying lengths. Notably, there is no evidence that specific sequence lengths are more prone to generating outliers. Instead, outliers are influenced by token-level semantics, consistently favoring the first tokens and semantically weak tokens, regardless of sequence length.

