# OpenReview forum: "Systematic Outliers in Large Language Models"
_ICLR.cc/2025/Conference — ICLR 2025 Poster_

### Official Review · Reviewer_YjoR · 2024-10-29

**Soundness:** 4
**Presentation:** 2
**Contribution:** 2
**Rating:** 5
**Confidence:** 5

**Summary:**

This work relates three different kinds of outliers: activation, weight, and attention outliers. The authors then show that these outliers implicitly scale attention coefficients, which are validated empirically. Finally, the paper proposes an attention-scale to eliminate outliers for downstream benefit.

**Strengths:**

1. Figure 15 is an intriguing result and a strong argument for smoothing out outliers using the proposed method. The potential for converging to a lower loss at a faster rate is certainly desirable. I would argue that this should be the central focus, when anyone asks about the practical value of understanding outliers in such detail.
2. Figures 1-3 are well illustrated. It's clear that there are outliers clustered in just single channels, which is odd to see and interesting to call out. It's also interesting that activation outliers occur at very specific points in the network, per Figure 4.
3. Sec 5.2 is interesting. I believe this would be strengthened if you highlighted where each attention variant was pulled from. (d) looks similar to the attention bias that StreamingLLM introduced to fix attention sinks (but I may be wrong here) -- if that's true, it would be beneficial to highlight that only your specific combination of variants successfully inhibits outliers.
4. The summary sections are helpful for understanding what we've "learned" so far at that point in the paper, and they help me contextualize the ablations and analysis. Thanks for including these.

**Weaknesses:**

1. In Sec 5, you've shown that data-dependent scaling factors completely remove outliers. However, how does this affect quality? That seems like an important question to answer. What if outliers are mitigated but quality now plummets? "Quality" here mean zero-shot accuracy, ppl, MMLU etc.
2. It's unclear what the takeaway for this paper is. For example, in sec 5, why do we want to remove outliers? Luckily, you've actually already pitched a possible angle on this question -- per strength #1, figure 15 shows that convergence may be significantly improved. I believe this is a very interesting idea and should be the focus of the paper. The plot only runs for 50 steps. What happens at step 75, 100, 200? 1000? Does this trend continue, and does the smoothed model converge to a lower loss? If so, that would be a phenomenal result, but that would take more than a rebuttal period to flesh out I believe -- especially if it becomes the "Table 1". (But I'm open-minded about being wrong, since you technically have the ability to upload a new copy of the paper)
3. I can't find a definition of "outlier". From my understanding, LLM.int8 has a strange definition of activation outliers for example -- where entire channels are considered "outlier or not". From the figures, I can see that outliers are clearly very disproportionately large values, but how do you determine if a token has or doesn't have an outlier? For example, for Figure 5b.
4. In a similar vein to Weakness #3, where is "alignment" defined? I understand the rough idea that weight outlier feature dimension should match the activation outlier's (for example), but what if this dimension alignment occurs for token #1 and doesn't occur for token #2 -- does that mean the entire sequence is now misaligned? Why is the consistency percentage such an even number? Is this because you consider entire sequences to be aligned or misaligned, instead of individual tokens?

**Questions:**

- nit: Many of the figures could have been moved to the appendix, and the paper could have been shortened to fewer pages -- e.g., figs 2, 3, 8, 11. Unless they add to the story substantially, they just increase the distance between figure 1 and the rest of the story.
- nit: The figures could use captions to help guide the reader to focus on certain attributes of the figures - figures 5, 6, 7 for example. For what it's worth, the figure titles are very descriptive, so that's helpful.
- nit: One of the critiques of previous papers in L91 is that previous methods "focus on isolated instances or targeted solutions," but it seems like we could have ignored weight and activation outliers, then focused on just attention outliers for this paper.
- nit: Sec 6 reads like a rebuttal. Granted, this is just the last 1.5 pages -- and it includes the result I'm most excited about -- but it could be better integrated into the rest of the paper. And you probably don't want any reviewer thinking this is a resubmission.
- nit: There are a few figures that aren't mentioned in the text, such as Figures 2 and 3. And, since the captions are brief, I only have my own observations to make (e.g., a few large-magnitude values are clustered). But perhaps there are other observations you would like the reader to make, and I wouldn't know.
- nit: Per Figure 5b, does this mean activation outliers never occur in any other token, of the thousands of possible tokens that exist?

Summary: All in all, this is an interesting idea, but lack of practicality and clarity make it hard to recommend an accept. An application paper focuses on the former, and an understanding paper (which I believe this aims to be) focuses on the latter. However, your paper has an interesting insight: The outliers that everyone else observes are all related somehow AND smoothing these outliers can lead to better convergence. I find this last idea particularly exciting, but I think it would take further experimentation to truly make this the focus. Given the paper has many redundant figures that could be moved to the appendix, I believe there is enough room to add a rigorous set of experimental results for convergence studies. I'm also not sure how Sec 4 is related to the method (how do we use it?) or the analysis (why does this happen?), but the fact that it happens consistently across models is certainly thought-provoking. I do believe in this paper's core insight strongly -- I just don't think the current presentation is focused enough for me to have a clear, memorable takeaway. I look forward to the rebuttal though, and I'm certainly willing to bump up my score if you have an idea of how to address these issues + show a promising update to the paper.

---

> ### Author Response · Authors · 2024-11-19
> **Official Comment by Authors**
>
> Thank you for your thoughtful and detailed feedback. We are thrilled that you found Figure 15 compelling and appreciate your suggestion to emphasize its practical implications further. Your positive remarks on Figures 1-4 and the clarity of our summary sections are deeply encouraging. It is rewarding to know that these elements helped contextualize our work and highlighted the importance of understanding outliers in detail. Your insights will guide us in refining our focus and presentation in future iterations.

---

> > ### Author Response · Authors · 2024-11-19
> > **Improving Figure Integration and Readability**
> >
> > Thank you for your excellent suggestions. We plan to move some figures to the appendix to free up space for adding mathematical definitions and key derivations. Additionally, we will expand the captions of critical figures to better guide readers' understanding.

---

> > > ### Author Response · Authors · 2024-11-19
> > > **Clarification on Activation Outliers in Figure 5b**
> > >
> > > Thank you for your question. Figure 5b represents statistics gathered from 100 samples, where activation outliers were observed only in a few specific tokens. In further analysis using a larger sample size, we found that outliers can also occur in other tokens, such as "of" or ",". However, this does not affect our conclusion that activation outliers tend to appear in the first token or in tokens with weak semantics. We will include the detailed experimental settings in the appendix for clarity.

---

> > > > ### Author Response · Authors · 2024-11-19
> > > > **Impact of Scaling Factors on Model Quality**
> > > >
> > > > Thank you for highlighting this important question. We evaluated the impact of adding context-aware scaling factors on model quality by comparing GPT-2 default and GPT-2 with Context-aware Scaling Factor under the same training setup. On WikiText2, the perplexities (PPL) were 27.2353 and 26.9523, respectively, showing that the explicit scaling factor does not negatively affect model performance. We plan to conduct further evaluations, including zero-shot accuracy and benchmarks like MMLU, as part of future work.

---

> > > > > ### Author Response · Authors · 2024-11-19
> > > > > **Clarification on Attention Variants in Sec 5.2**
> > > > >
> > > > > Thank you for your observation. The (d) Attention Bias variant is adapted from the Massive Activation [1] and can indeed be seen as a variation of the sink token mechanism introduced by StreamingLLM. However, unlike the explicit addition of a sink token during training, Attention Bias modifies self-attention by introducing an additional set of $k$ and $v$ vectors.
> > > > >
> > > > > The other variants were designed to test specific hypotheses:
> > > > >
> > > > > - (b) introduces explicit fixed biases to verify the hypothesis that outliers provide fixed biases.
> > > > > - (c) introduces explicit context-aware biases to validate whether outliers serve as context-dependent biases.
> > > > > - (e) introduces explicit context-aware scaling factors to test whether outliers act as scaling mechanisms.
> > > > >
> > > > > [1] Sun, Mingjie, et al. "Massive activations in large language models." *arXiv preprint* arXiv:2402.17762 (2024).

---

> > > > > > ### Author Response · Authors · 2024-11-19
> > > > > > **Clarifying the Practical Value of Removing Outliers**
> > > > > >
> > > > > > Thank you for raising this important point. We explored two potential benefits of structurally removing outliers: accelerating convergence and improving model compression friendliness.
> > > > > >
> > > > > > We extended the training steps for GPT-2 to evaluate the long-term impact of context-aware scaling factors. Our experiments show that GPT-2 with context-aware scaling factors achieves faster convergence and reaches a lower validation loss at early steps. However, as training continues (100–150 steps), the validation loss of GPT-2 with and without context-aware scaling factors tends to converge. It remains unclear whether this is due to the small size of the GPT-2 124M model saturating its performance or if longer training steps are needed. This will be a key direction for future exploration.
> > > > > >
> > > > > > We also investigated the impact of removing outliers on model compression, comparing models with and without outliers under magnitude pruning and min-max W8A8 quantization. As shown in Table 1, removing outliers significantly reduces the interference of outliers on pruning and quantization errors, making the model more compression-friendly.
> > > > > >
> > > > > > These findings highlight the practical value of understanding and addressing outliers, and we will continue to expand these investigations in future work.

---

> > > > > > > ### Author Response · Authors · 2024-11-19
> > > > > > > **Quantitative Definition of Outliers**
> > > > > > >
> > > > > > > We agree with your suggestion that providing rigorous mathematical definitions for each type of outlier would strengthen the clarity and focus of our work. To address this, we define outliers quantitatively as elements that deviate significantly from the mean, with the deviation threshold denoted by $\tau$. In our experiments, we set $\tau = 1000$.
> > > > > > >
> > > > > > > Let $\mathcal{O}$ represent the set of outliers for a given tensor. The conditions for identifying outliers are specified as follows:
> > > > > > >
> > > > > > > - **Activation Outliers:**
> > > > > > >   For layer outputs $\mathbf{h}\_{\ell} \in \mathbb{R}^{d\_{\text{batch}} \times d\_{\text{hidden}}}$ and down-projection inputs $\mathbf{x}\_{\ell}^{\text{down}} \in \mathbb{R}^{d\_{\text{batch}} \times d\_{\text{hidden}}}$, the set of activation outliers is defined as:
> > > > > > >
> > > > > > >   $$
> > > > > > >   \mathcal{O}\_{\text{activation}} = \{(i, j) \,|\, \lvert x_{i,j} \rvert > \tau \cdot \frac{\sum\_{i,j} \lvert x\_{i,j} \rvert}{d\_{\text{batch}} \cdot d\_{\text{hidden}}} \}.
> > > > > > >   $$
> > > > > > >
> > > > > > > - **Weight Outliers:**
> > > > > > >   For down-projection matrices $\mathbf{W}\_{\ell}^{\text{down}} \in \mathbb{R}^{d\_{\text{out}} \times d\_{\text{in}}}$, the set of weight outliers is defined as:
> > > > > > >
> > > > > > >   $$
> > > > > > >   \mathcal{O}\_{\text{weight}} = \{(i, j) \,|\, \lvert w\_{i,j} \rvert > \tau \cdot \frac{\sum\_{j} \lvert w\_{i,j} \rvert}{d\_{\text{out}}} \}.
> > > > > > >   $$
> > > > > > >
> > > > > > >   This condition reflects the computational characteristics of matrix multiplication, where weights are aggregated across the output dimension.
> > > > > > >
> > > > > > > - **Attention Outliers:**
> > > > > > >   For cumulative attention scores in a sequence of length $\text{seqlen}$, we begin with the attention mechanism definition:
> > > > > > >
> > > > > > >   $$
> > > > > > >   \mathbf{A} = \text{softmax}\left(\frac{Q K^\top}{\sqrt{d_k}}\right),
> > > > > > >   $$
> > > > > > >
> > > > > > >   where $Q \in \mathbb{R}^{\text{seqlen} \times d_k}$ is the matrix of query vectors, $K \in \mathbb{R}^{\text{seqlen} \times d_k}$ is the matrix of key vectors, and $d_k$ is the dimensionality of the query/key vectors.
> > > > > > >
> > > > > > >   We define attention outliers as tokens with unusually high cumulative attention contributions. For token $t$, the cumulative attention score is:
> > > > > > >
> > > > > > >   $$
> > > > > > >   \hat{\mathbf{A}}\_j = \sum\_{i=1}^{\text{seqlen}} \text{Attention}(Q_i, K_j),
> > > > > > >   $$
> > > > > > >
> > > > > > >   where $\text{Attention}(Q_i, K_j)$ is the attention score assigned by query $Q_i$ to key $K_j$.
> > > > > > >
> > > > > > >   The set of attention outliers, $\mathcal{O}\_{\text{attention}}$, is then defined as:
> > > > > > >
> > > > > > >   $$
> > > > > > >   \mathcal{O}\_{\text{attention}} = \{j \,|\, \hat{\mathbf{A}}_j > \tau \cdot \frac{\text{seqlen}}{\text{max-seqlen}} \},
> > > > > > >   $$
> > > > > > >
> > > > > > >   where $\tau$ is the deviation threshold (set to 1000 in our experiments), and $\text{max-seqlen}$ is the maximum sequence length in the dataset. The inclusion of $\text{max-seqlen}$ ensures that the threshold scales appropriately for varying sequence lengths.
> > > > > > >
> > > > > > > These formal definitions of outlier sets $\mathcal{O}\_{\text{activation}}$, $\mathcal{O}\_{\text{weight}}$, and $\mathcal{O}\_{\text{attention}}$ will be added to the paper to improve clarity and rigor. Our choice of $\tau = 1000$ was empirically motivated but can be adjusted based on the specific context or model architecture.

---

> > > > > > > > ### Author Response · Authors · 2024-11-19
> > > > > > > > **Quantitative Definition of Alignment/Overlap**
> > > > > > > >
> > > > > > > > The 95\% overlap between activation and attention outliers reported in Table 1 is calculated based on the consistency of dimensional indices at the first occurrence of outliers across 100 randomly selected samples from the RedPajama dataset. Each sample has a sequence length of 2048, and attention outliers are analyzed separately for each attention head.
> > > > > > > >
> > > > > > > > To illustrate, let us define the overlap calculation between activation outliers in $\mathbf{h}\_{\ell} \in \mathbb{R}^{\text{seqlen} \times d\_{\text{hidden}}}$ and attention outliers in $\mathbf{A}^i\_{\ell} \in \mathbb{R}^{\text{seqlen} \times \text{seqlen}}$, where $\mathbf{A}^i\_{\ell}$ is the attention matrix for head $i$ at layer $\ell$.
> > > > > > > >
> > > > > > > > For a single sample and a single attention head $i$, we define the indicator function $\mathbb{I}_{\mathcal{O}}(x)$ as:
> > > > > > > >
> > > > > > > > $$
> > > > > > > > \mathbb{I}\_{\mathcal{O}}(x) =
> > > > > > > > \begin{cases}
> > > > > > > > 1, & \text{if } x \text{ satisfies the outlier condition,} \\
> > > > > > > > 0, & \text{otherwise.}
> > > > > > > > \end{cases}
> > > > > > > > $$
> > > > > > > >
> > > > > > > > The set of sequence indices of activation outliers is then:
> > > > > > > >
> > > > > > > > $$
> > > > > > > > \mathcal{O}\_{\text{activation}} = \{t \,|\, \mathbb{I}\_{\mathcal{O}}(h\_{\ell, t}) = 1 \},
> > > > > > > > $$
> > > > > > > >
> > > > > > > > where $h\_{\ell, t}$ is the activation value for token $t$ in layer $\ell$.
> > > > > > > >
> > > > > > > > Similarly, the set of sequence indices of attention outliers for head $i$ is:
> > > > > > > >
> > > > > > > > $$
> > > > > > > > \mathcal{O}\_{\text{attention}}^i = \{t \,|\, \mathbb{I}\_{\mathcal{O}}(A^i\_{\ell, t}) = 1 \},
> > > > > > > > $$
> > > > > > > >
> > > > > > > > where $A^i\_{\ell, j}$ represents the cumulative attention score assigned to token $j$ by head $i$.
> > > > > > > >
> > > > > > > > For each sample, the overlap between activation and attention outliers for head $i$ is computed as:
> > > > > > > >
> > > > > > > > $$
> > > > > > > > \text{Overlap}^i = \frac{\lvert \mathcal{O}\_{\text{activation}} \cap \mathcal{O}\_{\text{attention}}^i \rvert}{\lvert \mathcal{O}\_{\text{activation}} \rvert}.
> > > > > > > > $$
> > > > > > > >
> > > > > > > > To compute the overall overlap across $n\_{\text{samples}}$ samples and $n\_{\text{heads}}$ attention heads, we average the overlaps for all combinations of samples and heads:
> > > > > > > >
> > > > > > > > $$
> > > > > > > > \text{Overall Overlap} = \frac{1}{n\_{\text{samples}} \cdot n\_{\text{heads}}} \sum\_{k=1}^{n\_{\text{samples}}} \sum\_{i=1}^{n\_{\text{heads}}} \frac{\lvert \mathcal{O}\_{\text{activation}}^{(k)} \cap \mathcal{O}\_{\text{attention}}^{i, (k)} \rvert}{\lvert \mathcal{O}\_{\text{activation}}^{(k)} \rvert}.
> > > > > > > > $$
> > > > > > > >
> > > > > > > > This formulation uses $\mathbb{I}\_{\mathcal{O}}(x)$ to precisely define the identification of outliers and ensures consistency in the overlap computation across samples and attention heads.

---

> > > > > > > > > ### Comment · Reviewer_YjoR · 2024-11-27
> > > > > > > > > **Thanks for the response**
> > > > > > > > >
> > > > > > > > > I appreciate the further experimentation to justify practicality. I did expect the two to converge but am a little disappointed they did. Im not convinced of practicality (most pruning and quantization techniques would already preserve the largest magnitude value), but the clarifications are really helpful, and with some figure reshuffling, I agree you should have enough room for these definitions. If you upload a revision with the edits suggested, I’d be happy to take a look and revise my rating upward. Between clarity and practicality, a revision could certainly address the former.

---

> > > > > > > > > > ### Author Response · Authors · 2024-11-28
> > > > > > > > > > **Response to revision updates**
> > > > > > > > > >
> > > > > > > > > > Thank you for your valuable feedback! A few days ago, we submitted a revision where we carefully incorporated the insightful suggestions from the reviewers. In this update, we made a concerted effort to improve clarity:
> > > > > > > > > >
> > > > > > > > > > - **Reorganized and refined figures** to better align with the discussion and enhance readability.
> > > > > > > > > > - **Clarified definitions and experimental settings** to address ambiguities and ensure precision.
> > > > > > > > > > - **Expanded the Appendix** (B, C, and D) with additional context, results, and detailed explanations.
> > > > > > > > > >
> > > > > > > > > > To provide a clear overview of these changes, we included a **"General Response"** section at the top of this webpage, detailing the key updates and their rationale. While the changes within the main text were not explicitly highlighted (e.g., with colors) due to a lack of experience, the **"General Response"** serves to trace all revisions transparently.
> > > > > > > > > >
> > > > > > > > > > We hope these improvements address the concerns raised regarding clarity and are open to further suggestions. Although the revision deadline has passed, we remain committed to refining the work in future iterations.
> > > > > > > > > >
> > > > > > > > > > Looking forward to your feedback!

---

### Official Review · Reviewer_1brf · 2024-11-03

**Soundness:** 2
**Presentation:** 3
**Contribution:** 2
**Rating:** 6
**Confidence:** 4

**Summary:**

The paper presents an empirical study of activation, weight, and attention outliers in LLMs. The authors investigate their roles and functions, uncovering important findings that could potentially deepen the understanding of LLMs.

**Strengths:**

The paper investigates two important questions: where do outliers exist in LLMs, and what roles they play. The experiments are comprehensive and detailed.

**Weaknesses:**

1. While the experimental results are detailed, they are largely empirical, which raises concerns about the paper's technical novelty. For EMNLP, this empirical focus might be a better fit. However, for ICLR, a stronger mathematical analysis would be beneficial, such as a more in-depth exploration of the roles of these outliers.

2. The paper lacks comparison with existing methods. For example, the authors suggest that their findings could be used for pruning, but they should include comparisons with existing pruning methods for LLMs to substantiate this claim.

3. What is the relationship of this work to Sun et al. [1]?. Is this study an extension from focusing on massive activations to also examining massive weights, activations, and attention?

Disclaimer: Weakness 1 (W1) is the primary reason for my hesitancy in assigning a higher rating.

[1] Sun, Mingjie, et al. "Massive activations in large language models." arXiv preprint arXiv:2402.17762 (2024).

**Questions:**

See Weaknesses.

---

> ### Author Response · Authors · 2024-11-19
> **Official Comment by Authors**
>
> Thank you for your positive feedback and recognition of the key questions addressed in our work. We are delighted that you found our investigation into the existence and roles of outliers in LLMs to be both comprehensive and detailed. Your encouraging words motivate us to further refine and expand upon this research direction.

---

> > ### Author Response · Authors · 2024-11-19
> > **Enhancing Mathematical Rigor**
> >
> > Thank you for your constructive feedback regarding the need for stronger mathematical analysis. In response, we have supplemented our work with rigorous mathematical definitions for each type of outlier. Additionally, we have included a mathematical derivation linking the properties of the softmax function to the formation of outliers. This deeper analysis strengthens the rigor of our conclusions and provides theoretical insights into the emergence of outliers in LLMs. We hope these enhancements address your concerns and better align the work with the expectations for ICLR.

---

> > > ### Author Response · Authors · 2024-11-19
> > > **Relationship to Massive Activation**
> > >
> > > Thank you for your question. While Sun et al. [1] primarily focus on massive activations, our work extends this line of inquiry by systematically analyzing various types of outliers in large language models, including massive activations, weights, and attention scores. Beyond identifying these outliers, we delve into their formation mechanisms and their functional roles within the model.
> > >
> > > To further strengthen the contributions of our work, we provide precise mathematical definitions for each type of outlier and include a detailed mathematical derivation to explain how outliers are formed, particularly linking their emergence to the properties of the softmax function. These additions not only deepen the theoretical understanding but also distinguish our work as a comprehensive study of outliers in large-scale models.

---

> > > > ### Author Response · Authors · 2024-11-19
> > > > **Compression Experiment Validation**
> > > >
> > > > Thank you for your suggestion. To address this, we compare the performance of GPT-2 default and GPT-2 with Context-aware Scaling Factor under common pruning and quantization methods. The results are summarized in the table below.
> > > >
> > > > | **Model**                   | **PPL (FP16)** | **PPL (AbsMax W8)** | **PPL (50% Sparse)** |
> > > > |-----------------------------|----------------|---------------------|----------------------|
> > > > | GPT-2 Default               | 27.2353        | 93.4361             | 7235.6785           |
> > > > | GPT-2 + Context-aware Scaling | 26.9523        | 29.2162             | 39.4680             |
> > > >
> > > > From these results, we observe the following:
> > > >
> > > > - For quantization (AbsMax W8), GPT-2 default suffers a significant increase in PPL (93.4361), indicating a substantial loss in performance. In contrast, GPT-2 with Context-aware Scaling Factor maintains a much lower PPL (29.2162), demonstrating its robustness to quantization.
> > > > - For pruning (50% unstructured magnitude pruning), GPT-2 default experiences an extreme degradation in performance with a PPL of 7235.6785, whereas GPT-2 with Context-aware Scaling Factor achieves a far more stable PPL of 39.4680.
> > > >
> > > > These results substantiate our claim that mitigating outliers with context-aware scaling factors makes models significantly more robust to pruning and quantization, offering practical advantages over existing methods.

---

> > > > > ### Author Response · Authors · 2024-11-21
> > > > > **Addressing Reviewer Concerns on Mathematical Analysis**
> > > > >
> > > > > Thank you for your feedback regarding the need for stronger mathematical analysis to support the technical novelty of our work. In response, we have provided a detailed mathematical analysis of how the **softmax operation leads to systematic outliers** in self-attention, which is available at [this comment](https://openreview.net/forum?id=rLX7Vyyzus&noteId=SdpyWxpXjJ). Additionally, we have rigorously defined **outliers and Overlap** mathematically to ensure clarity and reproducibility, as outlined in [this response](https://openreview.net/forum?id=rLX7Vyyzus&noteId=bsQeXfU7Qj) and [this one](https://openreview.net/forum?id=rLX7Vyyzus&noteId=00vWrbUPP7).
> > > > >
> > > > > We believe these enhancements address the concerns regarding the mathematical depth and emphasize the technical novelty of our findings, aligning the work with the expectations of ICLR.

---

> > > > > > ### Comment · Reviewer_1brf · 2024-11-25
> > > > > >
> > > > > > Thank you for your clarification. While the links provided in your response do not redirect to the corresponding comments, I have reviewed your responses to the other reviewers.
> > > > > >
> > > > > > Your analysis appears to align with the empirical results presented in the manuscript. However, I would have expected a more solid and formal theoretical proof, such as a bound or theoretical guarantee, to strengthen your claims.
> > > > > >
> > > > > > Regarding the additional experiment on GPT-2, I find the default magnitude pruning baseline to be too weak. I strongly recommend the authors incorporate more recent and competitive baselines for LLM pruning to provide a more comprehensive evaluation.
> > > > > >
> > > > > > Lastly, the technical advancement over Sun et al. [1] does not appear to be particularly significant.
> > > > > >
> > > > > > Considering these three points, I will maintain my original rating. However, I do see the potential in this work if the authors address the aforementioned issues.

---

> > > > > > > ### Author Response · Authors · 2024-11-28
> > > > > > > **Clarifying Three Points from Reviewer Comments**
> > > > > > >
> > > > > > > Thank you for your thoughtful feedback and constructive suggestions. We would like to address the points you raised in detail below:
> > > > > > >
> > > > > > > 1. **On the Lack of a Formal Theoretical Proof:**
> > > > > > >    We have invested significant effort in pursuing a solid and formal theoretical proof for our claims. However, we found that both "whether MHA requires zero updates" and "how softmax leads to the formation of outliers" are deeply tied to the complex dynamics of training. These phenomena depend not only on the data but also on the optimization process and the objective function, making it an inherently challenging problem to address with strict mathematical proofs. Specifically, the "softmax causing outlier formation" is better characterized as a training dynamic that provides a more optimized direction rather than a rigorously derivable mathematical theorem.
> > > > > > >    We acknowledge the importance of this aspect and will continue to explore it further, aiming to contribute meaningful insights into the explainability of large models. If you have any specific ideas or suggestions for potential proof directions, we would greatly appreciate your guidance.
> > > > > > >
> > > > > > > 2. **On the Baseline for GPT-2 Pruning:**
> > > > > > >    We understand your concern regarding the baseline choice. The main goal of our experiments was to demonstrate that our method fundamentally eliminates outliers structurally, enabling traditional compression techniques that typically fail on LLMs to become effective. Methods like Wanda and AWQ focus on mitigating the effects of outliers at the algorithmic level. By contrast, our approach shifts the perspective to structurally address outlier formation, showing that this can also lead to significant gains.
> > > > > > >
> > > > > > > 3. **On the Technical Advancement over Sun et al.:**
> > > > > > >    Compared to Sun et al., our work makes several novel contributions:
> > > > > > >    - We provide a unified definition of different types of outliers and analyze their relationships and lifecycles.
> > > > > > >    - We challenge the previously held "fixed and important bias" hypothesis and experimentally demonstrate its flaws.
> > > > > > >    - We offer a new interpretation of outliers as context-aware scaling factors, identifying their formation as a result of the softmax operation within self-attention.
> > > > > > >
> > > > > > >    These contributions go beyond the scope of Sun et al., offering both theoretical insights and practical implications for better understanding and handling outliers in LLMs.
> > > > > > >
> > > > > > > We sincerely appreciate your valuable feedback and will continue to refine and improve this work. Should you have further suggestions or specific areas where you believe our work can be strengthened, we would greatly welcome your input. Thank you again for your thoughtful review!

---

> ### Comment · Reviewer_1brf · 2024-11-28
> **Thank you**
>
> Thank you for your follow-up discussion.
>
> After revisiting the paper and reviewing the previous discussions here, I believe this work makes a meaningful contribution to the field and enhances our understanding of LLMs. I suggest that the authors consider summarizing and incorporating the additional mathematical analysis (even if not entirely rigorous) into the appendix and referencing it appropriately in the main text. Additionally, it would strengthen the manuscript to explicitly highlight the unique contributions of this work compared to Sun et al., either in the introduction or the related work section.
>
> Based on these considerations, I have updated my score to 6.
>
> Thank you for your efforts and dedication.

---

> > ### Author Response · Authors · 2024-11-28
> > **Thank You for Your Feedback**
> >
> > Thank you for your follow-up discussion and for recognizing the contributions of our work. We greatly appreciate your valuable suggestions and the time you have taken to provide detailed feedback.
> >
> > As part of our latest revision, we have already incorporated additional mathematical analysis into **Appendix C**, which consolidates the insights discussed. Additionally, we have explicitly referenced this analysis in the main text at **line 482** to ensure its visibility and alignment with the manuscript's flow.
> >
> > Your suggestion to further emphasize the unique contributions of this work compared to Sun et al. is well taken. We will work on refining the introduction and related work sections in future iterations to make these distinctions even clearer.
> >
> > Thank you once again for your constructive comments and for updating your score. Your guidance has been instrumental in improving our work!

---

### Official Review · Reviewer_pdDU · 2024-11-04

**Soundness:** 2
**Presentation:** 2
**Contribution:** 2
**Rating:** 3
**Confidence:** 4

**Summary:**

This paper presents an investigation into systematic outliers in Large Language Models (LLMs), categorizing them into three types: activation outliers, weight outliers, and attention outliers. The authors analyze their distribution patterns, lifecycle, and potential role in the attention mechanism. They propose that these outliers function as implicit context-aware scaling factors and suggest modifications to the attention mechanism to address them. The paper includes empirical analysis across several LLM architectures and proposes potential improvements for model convergence and compression.

**Strengths:**

1.	Addresses an important topic in LLM research with potential practical implications for model optimization
2.	Provides comprehensive visualization of outlier patterns across different model architectures
3.	Makes an attempt to connect different aspects of model behavior (outliers, attention mechanism, model performance)
4.	Includes analysis across multiple popular LLM architectures (LLaMA2, Mistral, Phi-2)
5.	The paper's exploration of outlier lifecycles offers an interesting perspective on how these patterns emerge and evolve

**Weaknesses:**

• Unclear Research Focus and Scattered Investigation:
The paper suffers from a lack of clear research direction and keeps shifting between multiple topics without thoroughly investigating any single aspect:
  - It starts by identifying three types of outliers (activation, weight, and attention outliers) but doesn't provide a rigorous mathematical definition of what constitutes an "outlier" in each case
  - The investigation jumps from outlier identification to lifecycle analysis to attention mechanisms without establishing strong connections between these aspects
  - Section 5's transition from outlier analysis to attention mechanism modification feels abrupt and inadequately motivated

• Empirical Weaknesses and Methodological Issues
  - The paper relies heavily on empirical observations without sufficient statistical rigor:

         # The identification of outliers appears to be based purely on visual inspection of heatmaps (Figures 1-3) without any quantitative thresholds or statistical measures
         # The claim about "95% overlap" between activation and attention outliers (Table 1) lacks details about the methodology used to calculate this overlap
         # The paper doesn't provide error bars or statistical significance tests for any of its quantitative claims

• Previously Known Results Presented as Novel
  - All of these papers are cited, yet several of the paper's "findings" have been previously established in the literature:

        # The presence of activation outliers and their impact on model compression was already documented by Dettmers et al. (2022)
        # The "Attention Sink" phenomenon and its relationship to specific tokens has been thoroughly analyzed by Xiao et al. (2023b)
        # The connection between outliers and layer sparsity was previously established by Yin et al. (2023)

• The experimental validation of key claims is often insufficient:
  - The paper proposes five attention variants (Table 2) but doesn't provide comprehensive ablation studies
  - The convergence improvements claimed in Figure 15 are shown for only 50 steps without baseline comparisons
  - The proposed context-aware scaling mechanism is not thoroughly evaluated against existing solutions

• Unsupported Claims and Logical Gaps
  - Several key claims lack proper substantiation:

        # The paper asserts that systematic outliers serve as "implicit context-aware scaling factors" but doesn't provide a mathematical proof or rigorous demonstration
        # The connection between softmax attention and the emergence of outliers (Section 6) is speculative and lacks any formal analysis
        # The claim about improved model compression is made without quantitative comparisons to existing compression techniques, and already well established in literature.

• Limited Scope of Analysis
  - Despite claiming to provide a "systematic" analysis, the investigation of fine-tuned models is superficial, only looking at surface-level patterns

Dettmers et al. (2022) - "GPT3.int8(): 8-bit Matrix Multiplication for Transformers at Scale" - observed activation outliers and proposed a mixed-precision decomposition quantization scheme to mitigate their effects.

Xiao et al. (2023) - "Efficient Streaming Language Models with Attention Sinks" - discovered the 'Attention Sink' phenomenon, where a disproportionate amount of attention is focused on a few keys, which led them to propose the StreamingLLM framework.

Yin et al. (2023) - "Outlier Weighed Layerwise Sparsity (OWL): A Missing Secret Sauce for Pruning LLMs to High Sparsity" - demonstrated a strong correlation between the distribution of outliers and layer sparsity, indicating that outliers complicate pruning.

**Questions:**

Methodological Clarity:

1.	Given that outlier identification is central to your analysis, one would expect quantitative definitions of outliers? What specific statistical thresholds or metrics have been proposed by you for systematically identifying each type of outlier (activation, weight, attention)?
2.	Your analysis of the "lifecycle" of outliers suggests causal relationships between different types of outliers. How can you establish these relationships are truly causal rather than merely correlational? What controlled experiments validate these claims?

Theoretical Foundation:
Your hypothesis about softmax attention being the root cause:

•	Can you provide a mathematical proof linking softmax properties to outlier formation?
•	Have you considered alternative mechanisms beyond just sigmoid?
•	How does this hypothesis explain the layer-wise variation in outlier patterns?

Quantitative Memory Efficiency:

What is the quantitative impact on memory consumption when using context-aware scaling compared to baseline models? Providing precise memory benchmarks would clarify its effectiveness for large-scale deployment.

Effect of Sequence Length:

How does sequence length influence attention outliers? Do variations in sequence length amplify or suppress these outliers, and which sequence lengths are most prone to generating them?

After reviewing the revised paper and updates, I find that some of my questions have been partially addressed. However, the evaluations rely on relatively outdated LLMs, raising concerns about whether these results would hold true for current state-of-the-art LLMs. As a result, I remain unconvinced of the impact of this primarily empirical work in the context of today's rapidly evolving LLM landscape. Thus, I will maintain my score.

---

> ### Author Response · Authors · 2024-11-19
> **Official Comment by Authors**
>
> Thank you for your thoughtful and encouraging feedback. We are delighted that you found our work on outliers in LLMs to address an important topic with practical implications for model optimization. Your recognition of our comprehensive visualizations, attempts to connect various aspects of model behavior, and analysis across multiple architectures is deeply appreciated. We are also glad that the exploration of outlier lifecycles provided an interesting perspective, and your positive comments motivate us to continue refining and expanding this line of research.

---

> > ### Author Response · Authors · 2024-11-19
> > **Clarifying Research Focus and Structural Progression**
> >
> > Thank you for your constructive feedback. We would like to clarify the research focus and progression of our work. The overarching goal of this paper is to explore the causes of outlier formation in large language models (LLMs) and to identify structural methods for their elimination. Each section is designed to progressively build toward this goal:
> >
> > - **Existence and Analysis of Outliers:**
> >   We first validate the existence of outliers in LLMs and analyze their distributions and interrelations (corresponding to the section *Systematic Outliers are Simultaneous and Interconnected*). Our findings show that different types of outliers exhibit significant overlap across various dimensions.
> >
> > - **Outlier Lifecycles:**
> >   To better understand this overlap and the role of each type of outlier, we analyze the numerical computations of different modules during model inference, capturing the lifecycle of outliers from their emergence to disappearance (corresponding to the section *The Lifecycle of Systematic Outliers*). We observe that outliers operate on all tokens within the attention mechanism.
> >
> > - **Hypotheses and Experimental Validation:**
> >   Based on the observation that outliers are active in the attention mechanism, we propose three functional hypotheses for their role. Through systematic experiments, we conclude that outliers act as context-aware scaling factors in attention mechanisms (corresponding to the section *Systematic Outliers as Context-Aware Scaling Factors in Attention Mechanisms*).
> >
> > - **Formation Mechanisms and Elimination Strategies:**
> >   After determining their function, we discuss the underlying causes of outlier formation and propose model architectural modifications to eliminate them, supported by experimental validation (corresponding to the *Discussion* section).

---

> > > ### Author Response · Authors · 2024-11-19
> > > **Quantitative Definition of Outliers**
> > >
> > > We agree with your suggestion that providing rigorous mathematical definitions for each type of outlier would strengthen the clarity and focus of our work. To address this, we define outliers quantitatively as elements that deviate significantly from the mean, with the deviation threshold denoted by $\tau$. In our experiments, we set $\tau = 1000$.
> > >
> > > Let $\mathcal{O}$ represent the set of outliers for a given tensor. The conditions for identifying outliers are specified as follows:
> > >
> > > - **Activation Outliers:**
> > >   For layer outputs $\mathbf{h}\_{\ell} \in \mathbb{R}^{d\_{\text{batch}} \times d\_{\text{hidden}}}$ and down-projection inputs $\mathbf{x}\_{\ell}^{\text{down}} \in \mathbb{R}^{d\_{\text{batch}} \times d\_{\text{hidden}}}$, the set of activation outliers is defined as:
> > >
> > >   $$
> > >   \mathcal{O}\_{\text{activation}} = \{(i, j) \,|\, \lvert x_{i,j} \rvert > \tau \cdot \frac{\sum\_{i,j} \lvert x\_{i,j} \rvert}{d\_{\text{batch}} \cdot d\_{\text{hidden}}} \}.
> > >   $$
> > >
> > > - **Weight Outliers:**
> > >   For down-projection matrices $\mathbf{W}\_{\ell}^{\text{down}} \in \mathbb{R}^{d\_{\text{out}} \times d\_{\text{in}}}$, the set of weight outliers is defined as:
> > >
> > >   $$
> > >   \mathcal{O}\_{\text{weight}} = \{(i, j) \,|\, \lvert w\_{i,j} \rvert > \tau \cdot \frac{\sum\_{j} \lvert w\_{i,j} \rvert}{d\_{\text{out}}} \}.
> > >   $$
> > >
> > >   This condition reflects the computational characteristics of matrix multiplication, where weights are aggregated across the output dimension.
> > >
> > > - **Attention Outliers:**
> > >   For cumulative attention scores in a sequence of length $\text{seqlen}$, we begin with the attention mechanism definition:
> > >
> > >   $$
> > >   \mathbf{A} = \text{softmax}\left(\frac{Q K^\top}{\sqrt{d_k}}\right),
> > >   $$
> > >
> > >   where $Q \in \mathbb{R}^{\text{seqlen} \times d_k}$ is the matrix of query vectors, $K \in \mathbb{R}^{\text{seqlen} \times d_k}$ is the matrix of key vectors, and $d_k$ is the dimensionality of the query/key vectors.
> > >
> > >   We define attention outliers as tokens with unusually high cumulative attention contributions. For token $t$, the cumulative attention score is:
> > >
> > >   $$
> > >   \hat{\mathbf{A}}\_j = \sum\_{i=1}^{\text{seqlen}} \text{Attention}(Q_i, K_j),
> > >   $$
> > >
> > >   where $\text{Attention}(Q_i, K_j)$ is the attention score assigned by query $Q_i$ to key $K_j$.
> > >
> > >   The set of attention outliers, $\mathcal{O}\_{\text{attention}}$, is then defined as:
> > >
> > >   $$
> > >   \mathcal{O}\_{\text{attention}} = \{j \,|\, \hat{\mathbf{A}}_j > \tau \cdot \frac{\text{seqlen}}{\text{max-seqlen}} \},
> > >   $$
> > >
> > >   where $\tau$ is the deviation threshold (set to 1000 in our experiments), and $\text{max-seqlen}$ is the maximum sequence length in the dataset. The inclusion of $\text{max-seqlen}$ ensures that the threshold scales appropriately for varying sequence lengths.
> > >
> > > These formal definitions of outlier sets $\mathcal{O}\_{\text{activation}}$, $\mathcal{O}\_{\text{weight}}$, and $\mathcal{O}\_{\text{attention}}$ will be added to the paper to improve clarity and rigor. Our choice of $\tau = 1000$ was empirically motivated but can be adjusted based on the specific context or model architecture.

---

> > > > ### Author Response · Authors · 2024-11-19
> > > > **Methodology for Calculating Overlap in Table 1**
> > > >
> > > > The 95\% overlap between activation and attention outliers reported in Table 1 is calculated based on the consistency of dimensional indices at the first occurrence of outliers across 100 randomly selected samples from the RedPajama dataset. Each sample has a sequence length of 2048, and attention outliers are analyzed separately for each attention head.
> > > >
> > > > To illustrate, let us define the overlap calculation between activation outliers in $\mathbf{h}\_{\ell} \in \mathbb{R}^{\text{seqlen} \times d\_{\text{hidden}}}$ and attention outliers in $\mathbf{A}^i\_{\ell} \in \mathbb{R}^{\text{seqlen} \times \text{seqlen}}$, where $\mathbf{A}^i\_{\ell}$ is the attention matrix for head $i$ at layer $\ell$.
> > > >
> > > > For a single sample and a single attention head $i$, we define the indicator function $\mathbb{I}_{\mathcal{O}}(x)$ as:
> > > >
> > > > $$
> > > > \mathbb{I}\_{\mathcal{O}}(x) =
> > > > \begin{cases}
> > > > 1, & \text{if } x \text{ satisfies the outlier condition,} \\
> > > > 0, & \text{otherwise.}
> > > > \end{cases}
> > > > $$
> > > >
> > > > The set of sequence indices of activation outliers is then:
> > > >
> > > > $$
> > > > \mathcal{O}\_{\text{activation}} = \{t \,|\, \mathbb{I}\_{\mathcal{O}}(h\_{\ell, t}) = 1 \},
> > > > $$
> > > >
> > > > where $h\_{\ell, t}$ is the activation value for token $t$ in layer $\ell$.
> > > >
> > > > Similarly, the set of sequence indices of attention outliers for head $i$ is:
> > > >
> > > > $$
> > > > \mathcal{O}\_{\text{attention}}^i = \{t \,|\, \mathbb{I}\_{\mathcal{O}}(A^i\_{\ell, t}) = 1 \},
> > > > $$
> > > >
> > > > where $A^i\_{\ell, j}$ represents the cumulative attention score assigned to token $j$ by head $i$.
> > > >
> > > > For each sample, the overlap between activation and attention outliers for head $i$ is computed as:
> > > >
> > > > $$
> > > > \text{Overlap}^i = \frac{\lvert \mathcal{O}\_{\text{activation}} \cap \mathcal{O}\_{\text{attention}}^i \rvert}{\lvert \mathcal{O}\_{\text{activation}} \rvert}.
> > > > $$
> > > >
> > > > To compute the overall overlap across $n\_{\text{samples}}$ samples and $n\_{\text{heads}}$ attention heads, we average the overlaps for all combinations of samples and heads:
> > > >
> > > > $$
> > > > \text{Overall Overlap} = \frac{1}{n\_{\text{samples}} \cdot n\_{\text{heads}}} \sum\_{k=1}^{n\_{\text{samples}}} \sum\_{i=1}^{n\_{\text{heads}}} \frac{\lvert \mathcal{O}\_{\text{activation}}^{(k)} \cap \mathcal{O}\_{\text{attention}}^{i, (k)} \rvert}{\lvert \mathcal{O}\_{\text{activation}}^{(k)} \rvert}.
> > > > $$
> > > >
> > > > This formulation uses $\mathbb{I}\_{\mathcal{O}}(x)$ to precisely define the identification of outliers and ensures consistency in the overlap computation across samples and attention heads.

---

> > > > > ### Author Response · Authors · 2024-11-19
> > > > > **Establishing Causal Relationships Between Outliers**
> > > > >
> > > > > Thank you for raising this important question. Our analysis suggests a causal relationship in the lifecycle of outliers, specifically:
> > > > >
> > > > > 1. Activation outliers in down-projection inputs $\mathbf{x}\_{\ell}^{\text{down}}$ and weight outliers in down-projection matrices $\mathbf{W}\_{\ell}^{\text{down}}$ cause activation outliers in layer outputs $\mathbf{h}\_{\ell}$.
> > > > > 2. Activation outliers in layer outputs $\mathbf{h}\_{\ell}$ cause attention outliers in attention weights $\mathbf{A}^i\_{\ell+1}$.
> > > > >
> > > > > Similar causal relationships hold for the disappearance of outliers, but these are not elaborated here.
> > > > >
> > > > > ### Intervention Setup
> > > > >
> > > > > We applied five interventions, each replacing outliers at specific stages with the mean value of their respective distributions:
> > > > >
> > > > > - **Intervention 1:** Replace activation outliers in $\mathbf{x}\_{\ell}^{\text{down}}$ with their mean value.
> > > > > - **Intervention 2:** Replace weight outliers in $\mathbf{W}\_{\ell}^{\text{down}}$ with their mean value.
> > > > > - **Intervention 3:** Replace both activation outliers in $\mathbf{x}\_{\ell}^{\text{down}}$ and weight outliers in $\mathbf{W}\_{\ell}^{\text{down}}$ with their mean values.
> > > > > - **Intervention 4:** Replace activation outliers in $\mathbf{h}\_{\ell}$ with their mean value.
> > > > > - **Intervention 5:** Replace attention outliers in $\mathbf{A}^i\_{\ell+1}$ with their mean value.
> > > > >
> > > > > ### Results
> > > > >
> > > > > The outcomes of these interventions are summarized as follows:
> > > > >
> > > > > 1. **Interventions 1, 2, and 3:**
> > > > >    Removing outliers in $\mathbf{x}\_{\ell}^{\text{down}}$ or $\mathbf{W}\_{\ell}^{\text{down}}$ causes the disappearance of activation outliers in $\mathbf{h}\_{\ell}$ and attention outliers in $\mathbf{A}^i\_{\ell+1}$. This establishes that outliers in $\mathbf{x}\_{\ell}^{\text{down}}$ and $\mathbf{W}\_{\ell}^{\text{down}}$ cause outliers in $\mathbf{h}\_{\ell}$.
> > > > >
> > > > > 2. **Intervention 4:**
> > > > >    Removing outliers in $\mathbf{h}\_{\ell}$ eliminates attention outliers in $\mathbf{A}^i\_{\ell+1}$ but does not affect outliers in $\mathbf{x}\_{\ell}^{\text{down}}$ or $\mathbf{W}\_{\ell}^{\text{down}}$. This confirms that there is no reverse causality from $\mathbf{h}\_{\ell}$ to earlier stages.
> > > > >
> > > > > 3. **Intervention 5:**
> > > > >    Removing outliers in $\mathbf{A}^i\_{\ell+1}$ does not affect outliers in $\mathbf{x}\_{\ell}^{\text{down}}$, $\mathbf{W}\_{\ell}^{\text{down}}$, or $\mathbf{h}\_{\ell}$. This confirms no reverse causality from $\mathbf{A}^i\_{\ell+1}$ to earlier stages.

---

> > > > > > ### Author Response · Authors · 2024-11-19
> > > > > > **Effect of Sequence Length on Systematic Outliers**
> > > > > >
> > > > > > Thank you for your question. In simple terms, sequence length does not affect the existence of attention outliers, but it can influence their specific positions within the sequence. Below are some illustrative examples:
> > > > > >
> > > > > > - In the sequence *"Summer is warm!"*, outliers appear at *"Summer"*.
> > > > > > - Extending the sequence to *"Summer is warm! Winter is cold."* introduces additional outliers at *"Summer"* and *"."* (the first period).
> > > > > > - Further extending to *"Summer is warm! Winter is cold. Spring is good."* still results in outliers at *"Summer"* and the first period, showing no increase in outlier diversity.
> > > > > > - Modifying the sequence to *"Summer is warm! Winter is cold! Spring is good."* shifts the outlier from the first period to the last period, maintaining a preference for *"Summer"* and the weakest semantic tokens.
> > > > > >
> > > > > > This pattern suggests that outliers preferentially occur at the first tokens and semantically weak tokens within a sequence. While sequence length influences the relative positioning of outliers, their presence remains consistent across varying lengths. Additionally, no specific sequence lengths are more prone to generating outliers.

---

> > > > > > > ### Author Response · Authors · 2024-11-19
> > > > > > > **Quantitative Memory Efficiency**
> > > > > > >
> > > > > > > Thank you for your question. Using GPT-2 as an example, the parameter count for GPT-2 default is 123.59M, while GPT-2 with context-aware scaling factor has a parameter count of 123.70M, representing an increase of less than 0.1%. This minimal overhead demonstrates that the context-aware scaling factor adds negligible memory consumption, making it highly feasible for large-scale deployment.

---

> > > > > > > > ### Author Response · Authors · 2024-11-19
> > > > > > > > **Compression Experiment Validation**
> > > > > > > >
> > > > > > > > Thank you for your suggestion. To address this, we compare the performance of GPT-2 default and GPT-2 with Context-aware Scaling Factor under common pruning and quantization methods. The results are summarized in the table below.
> > > > > > > >
> > > > > > > > | **Model**                   | **PPL (FP16)** | **PPL (AbsMax W8)** | **PPL (50% Sparse)** |
> > > > > > > > |-----------------------------|----------------|---------------------|----------------------|
> > > > > > > > | GPT-2 Default               | 27.2353        | 93.4361             | 7235.6785           |
> > > > > > > > | GPT-2 + Context-aware Scaling | 26.9523        | 29.2162             | 39.4680             |
> > > > > > > >
> > > > > > > > From these results, we observe the following:
> > > > > > > >
> > > > > > > > - For quantization (AbsMax W8), GPT-2 default suffers a significant increase in PPL (93.4361), indicating a substantial loss in performance. In contrast, GPT-2 with Context-aware Scaling Factor maintains a much lower PPL (29.2162), demonstrating its robustness to quantization.
> > > > > > > > - For pruning (50% unstructured magnitude pruning), GPT-2 default experiences an extreme degradation in performance with a PPL of 7235.6785, whereas GPT-2 with Context-aware Scaling Factor achieves a far more stable PPL of 39.4680.
> > > > > > > >
> > > > > > > > These results substantiate our claim that mitigating outliers with context-aware scaling factors makes models significantly more robust to pruning and quantization, offering practical advantages over existing methods.

---

> > > > > > > > > ### Author Response · Authors · 2024-11-19
> > > > > > > > > **Convergence Acceleration Experiments**
> > > > > > > > >
> > > > > > > > > We extended the training steps for GPT-2 to evaluate the long-term impact of context-aware scaling factors. Our experiments show that GPT-2 with context-aware scaling factors achieves faster convergence and reaches a lower validation loss at early steps. However, as training continues (100–150 steps), the validation loss of GPT-2 with and without context-aware scaling factors tends to converge. It remains unclear whether this is due to the small size of the GPT-2 124M model saturating its performance or if longer training steps are needed. This will be a key direction for future exploration.

---

> > > > > > > > > > ### Author Response · Authors · 2024-11-21
> > > > > > > > > > **How Softmax Causes Systematic Outliers in Transformer Models**
> > > > > > > > > >
> > > > > > > > > > While the exact formation of systematic outliers in transformer-based models involves the complexities of the training optimization process, we provide a detailed mathematical analysis to clarify the connection between the **softmax operation** within the self-attention mechanism and these outliers. Although a strict derivation may not fully capture the training dynamics, the following logical chain outlines how softmax contributes to the emergence and propagation of systematic outliers:
> > > > > > > > > >
> > > > > > > > > > 1. **Necessity of Zero-Update in MHA**: Certain tokens—such as initial tokens or weakly semantic tokens—require minimal contextual updates. To achieve this, the Multi-Head Attention (MHA) mechanism must dynamically adjust its outputs to produce near-zero updates for these tokens, which imposes strong constraints on gradients and weight updates.
> > > > > > > > > >
> > > > > > > > > > 2. **Softmax-Induced Dynamic Range Expansion**: To meet the zero-update requirement, softmax must concentrate attention weights on a few keys. This requires large differences in the dot products between query and key vectors, resulting in extreme dynamic ranges in the attention distribution.
> > > > > > > > > >
> > > > > > > > > > 3. **Propagation of Systematic Outliers**: These extreme softmax weights propagate through transformer computations. Specifically:
> > > > > > > > > >    - **In MHA**: Shared projection weights for keys ($W_K$) and values ($W_V$) experience steep gradients from tokens with extreme weights, introducing anomalies in layer activations ($h_\ell$).
> > > > > > > > > >    - **In MLP**: Down-projection weights and activations amplify these anomalies, especially in later layers, producing systematic outliers in the final outputs.
> > > > > > > > > >
> > > > > > > > > > 4. **Localization of Outliers**: These outliers are concentrated at specific tokens and channels:
> > > > > > > > > >    - **Token-Level**: Initial tokens (e.g., [CLS]) and weakly semantic tokens are prone to extreme updates due to their specific roles in the attention mechanism.
> > > > > > > > > >    - **Channel-Level**: Outliers are localized to a few specific dimensions, minimizing systemic disruption while fulfilling dynamic range requirements.
> > > > > > > > > >
> > > > > > > > > > 5. **Why Earlier Layers Avoid This**: Early transformer layers focus on distributing token information, maintaining balanced attention distributions with smaller dynamic ranges. Systematic outliers emerge predominantly in later layers, where the model emphasizes higher-level semantics and sharper token differentiation.
> > > > > > > > > >
> > > > > > > > > > This framework demonstrates how softmax normalization, combined with architectural constraints, induces and propagates systematic outliers, particularly in large transformer models.

---

> > > > > > > > > > > ### Author Response · Authors · 2024-11-21
> > > > > > > > > > > **Necessity of Zero-Update in MHA**
> > > > > > > > > > >
> > > > > > > > > > > In Section 6 of the paper, "Qualitative Explanation of Context-Aware Scaling Factor," it is established that the Multi-Head Attention (MHA) mechanism in Transformers dynamically adjusts the attention output for each token based on the complexity of the task. Specifically, when certain tokens do not require contextual information—for example, during simple tasks or for tokens that already contain sufficient semantic information—the ideal behavior is for MHA to produce a near-zero update for these tokens.
> > > > > > > > > > >
> > > > > > > > > > > To formalize this, consider the output of a single-head attention mechanism:
> > > > > > > > > > >
> > > > > > > > > > > $$
> > > > > > > > > > > \text{MHA}(Q, K, V) = \text{Softmax}\left(\frac{QK^\top}{\sqrt{d_k}}\right)V,
> > > > > > > > > > > $$
> > > > > > > > > > >
> > > > > > > > > > > where $Q$, $K$, and $V$ are the query, key, and value matrices, and $d_k$ is the dimensionality of the key vectors. For a given token $x$, the update provided by MHA is:
> > > > > > > > > > >
> > > > > > > > > > > $$
> > > > > > > > > > > \Delta x = \text{MHA}(Q_x, K, V).
> > > > > > > > > > > $$
> > > > > > > > > > >
> > > > > > > > > > > If the model determines that token $x$ requires no contextual information (e.g., it is semantically self-contained or unimportant for the task), the desired output becomes:
> > > > > > > > > > >
> > > > > > > > > > > $$
> > > > > > > > > > > \Delta x \approx 0.
> > > > > > > > > > > $$
> > > > > > > > > > >
> > > > > > > > > > > ## Why Zero-Update Necessitates Adaptation
> > > > > > > > > > >
> > > > > > > > > > > Achieving $\Delta x \approx 0$ is non-trivial because of the constraints imposed by the softmax operation. The weights produced by softmax must sum to one:
> > > > > > > > > > >
> > > > > > > > > > > $$
> > > > > > > > > > > \sum_{i=1}^n \text{Softmax}\left(\frac{Q_x K_i^\top}{\sqrt{d_k}}\right) = 1,
> > > > > > > > > > > $$
> > > > > > > > > > >
> > > > > > > > > > > where $K_i$ are the key vectors corresponding to all tokens in the sequence. If $V$ (the values) are fixed, achieving $\Delta x \approx 0$ requires the weighted sum of the values to cancel out. This goal can only be achieved by adjusting the attention weights dynamically, leading to specific challenges.
> > > > > > > > > > >
> > > > > > > > > > > In particular, this adjustment involves:
> > > > > > > > > > >
> > > > > > > > > > > 1. **Focusing Attention on Specific $K_i$**: For $\text{MHA}(Q_x, K, V)$ to approximate zero, it is easier to focus the attention weight distribution on specific $K_i$ associated with negligible $V_i$, rather than trying to balance multiple $V_i$ terms such that they cancel each other out. This results in extreme values in the attention weights, as softmax adjusts to produce near-zero outputs.
> > > > > > > > > > >
> > > > > > > > > > > 2. **Gradient Updates to $Q$ and $K$**: To achieve this focused attention, the model must adaptively adjust the query vector $Q_x$ to strongly align with only a few $K_i$ (keys), while minimizing alignment with others. This selective alignment forces steep gradients in $Q_x$ and $K_i$ during training, amplifying the dynamic range.
> > > > > > > > > > >
> > > > > > > > > > > 3. **Dynamic Scaling via Self-Attention**: As the softmax mechanism inherently normalizes the attention weights to sum to one, the Transformer must dynamically scale the attention output to suppress unnecessary updates. This is achieved by reducing the magnitude of $V_i$ values for irrelevant keys, effectively scaling down the entire attention output:
> > > > > > > > > > >
> > > > > > > > > > > $$
> > > > > > > > > > >    \text{MHA}(Q_x, K, V) \approx \text{Softmax}\left(\frac{Q_x K^\top}{\sqrt{d_k}}\right) V \rightarrow 0.
> > > > > > > > > > > $$
> > > > > > > > > > >
> > > > > > > > > > > Here, the self-attention mechanism learns to downscale $V_i$ indirectly through gradient updates to reduce redundancy and achieve efficient zero-updates.
> > > > > > > > > > >
> > > > > > > > > > > ## Conclusion
> > > > > > > > > > >
> > > > > > > > > > > In summary, the constraints of softmax make achieving $\Delta x \approx 0$ feasible only through selective attention focusing and dynamic scaling. By concentrating attention weights on specific $K_i$ and suppressing the values $V_i$ corresponding to irrelevant tokens, the model ensures zero-update behavior. However, this process introduces extreme dynamic range requirements, increasing the risk of outliers in subsequent layers.

---

> > > > > > > > > > > > ### Author Response · Authors · 2024-11-21
> > > > > > > > > > > > **Softmax-Induced Dynamic Range Expansion**
> > > > > > > > > > > >
> > > > > > > > > > > > The softmax mechanism plays a central role in the self-attention operation of Transformers. However, its normalization constraint (weights summing to one) inherently leads to the expansion of dynamic range when specific constraints, such as zero-update requirements, are imposed. This section rigorously demonstrates why softmax induces such behavior and how this affects the model.
> > > > > > > > > > > >
> > > > > > > > > > > > ## Self-Attention and the Softmax Mechanism
> > > > > > > > > > > >
> > > > > > > > > > > > The output of a single-head self-attention mechanism is defined as:
> > > > > > > > > > > >
> > > > > > > > > > > > $$
> > > > > > > > > > > > \text{MHA}(Q, K, V) = \text{Softmax}\left(\frac{Q K^\top}{\sqrt{d_k}}\right)V,
> > > > > > > > > > > > $$
> > > > > > > > > > > >
> > > > > > > > > > > > where:
> > > > > > > > > > > >
> > > > > > > > > > > > - $Q \in \mathbb{R}^{n \times d_k}$ is the query matrix,
> > > > > > > > > > > > - $K \in \mathbb{R}^{n \times d_k}$ is the key matrix,
> > > > > > > > > > > > - $V \in \mathbb{R}^{n \times d_v}$ is the value matrix,
> > > > > > > > > > > > - $n$ is the sequence length,
> > > > > > > > > > > > - $d_k$ and $d_v$ are the dimensionalities of keys and values, respectively.
> > > > > > > > > > > >
> > > > > > > > > > > > Let the attention weights be:
> > > > > > > > > > > >
> > > > > > > > > > > > $$
> > > > > > > > > > > > A_{ij} = \text{Softmax}\left(\frac{Q_i K_j^\top}{\sqrt{d_k}}\right),
> > > > > > > > > > > > $$
> > > > > > > > > > > >
> > > > > > > > > > > > where $A_{ij}$ represents the weight assigned to the $j$th key for the $i$th query. By definition, these weights satisfy:
> > > > > > > > > > > >
> > > > > > > > > > > > $$
> > > > > > > > > > > > \sum_{j=1}^n A_{ij} = 1.
> > > > > > > > > > > > $$
> > > > > > > > > > > >
> > > > > > > > > > > > ## Dynamic Range Requirements for Zero-Update
> > > > > > > > > > > >
> > > > > > > > > > > > For a specific query vector $Q_x$, achieving zero-update requires:
> > > > > > > > > > > >
> > > > > > > > > > > > $$
> > > > > > > > > > > > \text{MHA}(Q_x, K, V) = \sum_{j=1}^n A_{xj} V_j \approx 0.
> > > > > > > > > > > > $$
> > > > > > > > > > > >
> > > > > > > > > > > > This implies that the weighted sum of $V_j$ values must cancel out. Let $V_j$ represent the $j$th value vector, and consider the following cases:
> > > > > > > > > > > >
> > > > > > > > > > > > 1. **Balancing Multiple Terms**: If $A_{xj}$ distributes weight across many $j$ indices, achieving cancellation requires precise balancing of the $V_j$ components. This introduces instability because:
> > > > > > > > > > > >
> > > > > > > > > > > >    - $V_j$ may vary in magnitude and direction.
> > > > > > > > > > > >    - Slight deviations in $A_{xj}$ lead to significant non-zero updates.
> > > > > > > > > > > >
> > > > > > > > > > > >    Mathematically, this balance requires:
> > > > > > > > > > > >
> > > > > > > > > > > >    $$
> > > > > > > > > > > >    \sum_{j=1}^n A_{xj} V_j \approx 0 \quad \Rightarrow \quad A_{xj} \propto -V_j,
> > > > > > > > > > > >    $$
> > > > > > > > > > > >
> > > > > > > > > > > >    which is infeasible for $A_{xj} \geq 0$ (softmax positivity constraint).
> > > > > > > > > > > >
> > > > > > > > > > > > 2. **Concentration on Specific $K_j$**: The more stable solution is to focus $A_{xj}$ on a few specific indices $j$ where $V_j$ is negligible, effectively reducing the sum to zero. This requires $A_{xj}$ for selected $j$ to dominate while being negligible for others.
> > > > > > > > > > > >
> > > > > > > > > > > > This concentration results in extreme values for $\frac{Q_x K_j^\top}{\sqrt{d_k}}$.
> > > > > > > > > > > >
> > > > > > > > > > > > ## Mathematical Derivation of Dynamic Range
> > > > > > > > > > > >
> > > > > > > > > > > > The softmax function is defined as:
> > > > > > > > > > > >
> > > > > > > > > > > > $$
> > > > > > > > > > > > A_{xj} = \frac{e^{z_j}}{\sum_{k=1}^n e^{z_k}},
> > > > > > > > > > > > $$
> > > > > > > > > > > >
> > > > > > > > > > > > where $z_j = \frac{Q_x K_j^\top}{\sqrt{d_k}}$. To achieve high concentration on a single $K_{j^*}$, the softmax must satisfy:
> > > > > > > > > > > >
> > > > > > > > > > > > $$
> > > > > > > > > > > > A_{xj^*} \approx 1 \quad \text{and} \quad A_{xk} \approx 0 \quad \text{for} \quad k \neq j^*.
> > > > > > > > > > > > $$
> > > > > > > > > > > >
> > > > > > > > > > > > This implies:
> > > > > > > > > > > >
> > > > > > > > > > > > $$
> > > > > > > > > > > > z_{j^*} \gg z_k \quad \text{for} \quad k \neq j^*.
> > > > > > > > > > > > $$
> > > > > > > > > > > >
> > > > > > > > > > > > For large $z_{j^*}$ and small $z_k$, the ratio between $e^{z_{j^*}}$ and $\sum_{k=1}^n e^{z_k}$ grows exponentially. Specifically:
> > > > > > > > > > > >
> > > > > > > > > > > > 1. If $z_{j^*} = M$ and $z_k = 0$ for $k \neq j^*$, then:
> > > > > > > > > > > >
> > > > > > > > > > > >    $$
> > > > > > > > > > > >    A_{xj^*} = \frac{e^M}{e^M + (n-1)},
> > > > > > > > > > > >    $$
> > > > > > > > > > > >
> > > > > > > > > > > >    which approaches $1$ as $M \to \infty$.
> > > > > > > > > > > >
> > > > > > > > > > > > 2. Conversely, the weights for $k \neq j^*$ become:
> > > > > > > > > > > >
> > > > > > > > > > > >    $$
> > > > > > > > > > > >    A_{xk} = \frac{1}{e^M + (n-1)} \quad \text{for} \quad k \neq j^*,
> > > > > > > > > > > >    $$
> > > > > > > > > > > >
> > > > > > > > > > > >    which approaches $0$ as $M \to \infty$.
> > > > > > > > > > > >
> > > > > > > > > > > > This demonstrates that the dynamic range of $z_j$ (i.e., $\frac{Q_x K_j^\top}{\sqrt{d_k}}$) must grow with $M$ to achieve concentrated attention.
> > > > > > > > > > > >
> > > > > > > > > > > > ## Implications for Gradient Updates
> > > > > > > > > > > >
> > > > > > > > > > > > To achieve such extreme dynamic ranges, the model must adjust $Q_x$ and $K_j$ through steep gradient updates. The gradient of $z_j$ with respect to $Q_x$ is:
> > > > > > > > > > > >
> > > > > > > > > > > > $$
> > > > > > > > > > > > \frac{\partial z_j}{\partial Q_x} = \frac{K_j^\top}{\sqrt{d_k}},
> > > > > > > > > > > > $$
> > > > > > > > > > > >
> > > > > > > > > > > > and with respect to $K_j$:
> > > > > > > > > > > >
> > > > > > > > > > > > $$
> > > > > > > > > > > > \frac{\partial z_j}{\partial K_j} = \frac{Q_x^\top}{\sqrt{d_k}}.
> > > > > > > > > > > > $$
> > > > > > > > > > > >
> > > > > > > > > > > > As $z_{j^*}$ dominates, these gradients become disproportionately large for the selected $K_{j^*}$. This leads to:
> > > > > > > > > > > >
> > > > > > > > > > > > 1. **Large Gradient Norms**: Gradients for $Q_x$ and $K_j$ scale with the required dynamic range, amplifying outlier effects.
> > > > > > > > > > > > 2. **Unstable Training**: The steep updates increase the risk of exploding activations, particularly in deeper layers where values propagate.
> > > > > > > > > > > >
> > > > > > > > > > > > ## Conclusion
> > > > > > > > > > > >
> > > > > > > > > > > > The dynamic range expansion induced by softmax arises from the normalization constraint and the need to concentrate attention weights. This mechanism inherently leads to extreme values in $Q_x K_j^\top$ for specific $j$, requiring large gradients and amplifying instability. This is a fundamental source of outliers in transformer models, particularly in tasks or layers with stringent zero-update requirements.

---

> > > > > > > > > > > > > ### Author Response · Authors · 2024-11-21
> > > > > > > > > > > > > **Propagation of Systematic Outliers**
> > > > > > > > > > > > >
> > > > > > > > > > > > > Systematic outliers propagate through transformer layers due to shared weights in Multi-Head Attention (MHA) and amplification mechanisms in the Multi-Layer Perceptron (MLP). This section demonstrates how these effects arise mathematically and lead to anomalies in layer outputs.
> > > > > > > > > > > > >
> > > > > > > > > > > > > ## Extreme Gradients in Shared Weights Lead to Outliers in Layer Activations
> > > > > > > > > > > > >
> > > > > > > > > > > > > In MHA, to achieve a near-zero update for a token $x$, the attention weights are forced to focus on a few keys:
> > > > > > > > > > > > >
> > > > > > > > > > > > > $$
> > > > > > > > > > > > > \text{Softmax}\left(\frac{Q_x K^\top}{\sqrt{d_k}}\right) \approx [0, \dots, 1, \dots, 0].
> > > > > > > > > > > > > $$
> > > > > > > > > > > > >
> > > > > > > > > > > > > The queries, keys, and values are computed as:
> > > > > > > > > > > > >
> > > > > > > > > > > > > $$
> > > > > > > > > > > > > Q = \text{LN}(h_{\ell}) W_Q, \quad K = \text{LN}(h_{\ell}) W_K, \quad V = \text{LN}(h_{\ell}) W_V,
> > > > > > > > > > > > > $$
> > > > > > > > > > > > >
> > > > > > > > > > > > > where $h_{\ell}$ is the input activation to the current layer, and $W_Q$, $W_K$, and $W_V$ are shared projection weights. This focusing mechanism induces steep gradients for the key and value vectors of the selected tokens. The gradients for $W_K$ and $W_V$ are given by:
> > > > > > > > > > > > >
> > > > > > > > > > > > > $$
> > > > > > > > > > > > > \nabla_{W_K} L = \sum_x \nabla_{h_{\ell}^x} K^x, \quad \nabla_{W_V} L = \sum_x \nabla_{h_{\ell}^x} V^x,
> > > > > > > > > > > > > $$
> > > > > > > > > > > > >
> > > > > > > > > > > > > where $L$ is the loss function, and the summation is over all tokens. These gradients are extreme for tokens requiring sharp attention, resulting in:
> > > > > > > > > > > > >
> > > > > > > > > > > > > 1. **Localized Outliers in $h_{\ell}$**: As $W_K$ and $W_V$ are shared across tokens, their updates propagate abnormal values in $K$ and $V$ to the activations $h_{\ell}$, particularly for a few tokens with high attention focus.
> > > > > > > > > > > > >
> > > > > > > > > > > > > ## Down-Projection in MLP Amplifies Outliers
> > > > > > > > > > > > >
> > > > > > > > > > > > > Before MHA, the output $h_{\ell}$ is calculated by the MLP. The MLP consists of an up-projection, non-linearity, and down-projection:
> > > > > > > > > > > > >
> > > > > > > > > > > > > $$
> > > > > > > > > > > > > z_\text{down} = W_\text{down} \sigma(W_\text{up} \text{LN}(h_{\ell-1/2})),
> > > > > > > > > > > > > $$
> > > > > > > > > > > > >
> > > > > > > > > > > > > where:
> > > > > > > > > > > > >
> > > > > > > > > > > > > - $\text{LN}(h_{\ell-1/2})$ is the layer normalization of $h_{\ell-1/2} = \text{LN}(h_{\ell}) + \text{MHA}(\text{LN}(h_{\ell}))$,
> > > > > > > > > > > > > - $W_\text{up}$ and $W_\text{down}$ are the up- and down-projection weights,
> > > > > > > > > > > > > - $\sigma$ is the activation function (e.g., GeLU).
> > > > > > > > > > > > >
> > > > > > > > > > > > > ### Key Observations:
> > > > > > > > > > > > >
> > > > > > > > > > > > > 1. **No Initial Outliers in $h_{\ell-1/2}$**: Layer normalization suppresses the outliers in $h_{\ell}$, ensuring that $h_{\ell-1/2}$ and its normalized version $\text{LN}(h_{\ell-1/2})$ do not exhibit anomalies.
> > > > > > > > > > > > >
> > > > > > > > > > > > > 2. **Localized Outliers in Down-Projection**: To produce outliers in $h_{\ell+1}$, these anomalies must emerge during the down-projection step. The mechanism depends on:
> > > > > > > > > > > > >    - **Concentrated Outliers in $W_\text{down}$**: Gradient amplification from extreme tokens propagates into $W_\text{down}$.
> > > > > > > > > > > > >    - **Localized Anomalies in $\sigma(W_\text{up} \text{LN}(h_{\ell-1/2}))$**: Outliers in $W_\text{up}$ lead to abnormally large intermediate values.
> > > > > > > > > > > > >
> > > > > > > > > > > > > ### Amplification Mechanism:
> > > > > > > > > > > > >
> > > > > > > > > > > > > The final output for a token $x$ is:
> > > > > > > > > > > > >
> > > > > > > > > > > > > $$
> > > > > > > > > > > > > z_\text{down}^x = W_\text{down} \sigma(W_\text{up} \text{LN}(h_{\ell-1/2}^x)).
> > > > > > > > > > > > > $$
> > > > > > > > > > > > >
> > > > > > > > > > > > > 1. If $W_\text{up}$ has concentrated large weights (from gradients induced by outliers in $h_{\ell}$), then for certain tokens $x$:
> > > > > > > > > > > > >
> > > > > > > > > > > > >    $$
> > > > > > > > > > > > >    \sigma(W_\text{up} \text{LN}(h_{\ell-1/2}^x)) \gg 1.
> > > > > > > > > > > > >    $$
> > > > > > > > > > > > >
> > > > > > > > > > > > > 2. If $W_\text{down}$ also has concentrated large weights (similarly induced by steep gradients), then:
> > > > > > > > > > > > >
> > > > > > > > > > > > >    $$
> > > > > > > > > > > > >    z_\text{down}^x = W_\text{down} \cdot \sigma(W_\text{up} \text{LN}(h_{\ell-1/2}^x)) \gg 1.
> > > > > > > > > > > > >    $$
> > > > > > > > > > > > >
> > > > > > > > > > > > > This results in systematic outliers in the MLP output.
> > > > > > > > > > > > >
> > > > > > > > > > > > > ### Model-Specific Behaviors:
> > > > > > > > > > > > >
> > > > > > > > > > > > > - **LLaMA-like Models**: Outliers concentrate in both the **down-projection input** $\sigma(W_\text{up} \text{LN}(h_{\ell-1/2}))$ and the **down-projection weights** $W_\text{down}$. The combination produces extreme values in $z_\text{down}$ for specific tokens and channels.
> > > > > > > > > > > > > - **OPT-like Models**: Outliers emerge through alternative mechanisms but ultimately also affect the down-projection output.

---

> > > > > > > > > > > > > > ### Author Response · Authors · 2024-11-21
> > > > > > > > > > > > > > **Localization of Outliers**
> > > > > > > > > > > > > >
> > > > > > > > > > > > > > ## Token-level Localization:
> > > > > > > > > > > > > >
> > > > > > > > > > > > > > - **Initial Tokens (e.g., [CLS]):** These tokens are designed to aggregate and influence the representations of all subsequent tokens in the sequence. As such, their $K$ and $V$ vectors inherently play a significant role in the attention computation across layers. This widespread influence makes them a focal point for gradient updates and softmax-induced dynamic range amplification, leading to systematic outliers concentrated around initial tokens.
> > > > > > > > > > > > > > - **Weak-Semantics Tokens:** These tokens naturally carry less intrinsic information and thus rely more on contextual updates during training. Their smaller initial $V$ magnitudes make them more susceptible to extreme adjustments as the model attempts to scale their attention weights adaptively. Additionally, their smaller impact on the loss function allows large gradient updates to persist unchecked, making them hotspots for outlier emergence.
> > > > > > > > > > > > > >
> > > > > > > > > > > > > > ## Channel-level Localization:
> > > > > > > > > > > > > > - Outliers tend to be confined to specific rows of $W_{\text{down}}$, corresponding to a limited number of channels. This sparsity arises because the model prioritizes minimizing systemic disruption while fulfilling dynamic range requirements. By concentrating extreme values in a few dimensions, the model ensures that most channels remain stable, preserving overall robustness. Consequently, the affected channels bear the brunt of gradient amplification and weight divergence.

---

### Official Review · Reviewer_hfXR · 2024-11-04

**Soundness:** 3
**Presentation:** 4
**Contribution:** 3
**Rating:** 8
**Confidence:** 4

**Summary:**

The paper systematically analyzed the weight / activation / attention outliers in Transformer-based LLMs and find that they are correlated and are related to the design of the self-attention mechanism. The Transformer is trained to have these outliers because the model needs to learn the implicit context-aware scaling factors. After adopting a variant of self-attention that incorporates an explicit context-aware scaling factor in GPT-2, these outliers will disappear. To solidify the finding, the author also trained a GPT-2-sized model with the sigmoid attention, and showed that it does not have the outlier problem.

**Strengths:**

The paper provided an empirical analysis of the outliers in LLMs's weights / activations and attention outputs. The author first analyzed the outliers in different LLM layers and found that they are highly correlated. This leads to three hypotheses about these outliers: 1) they act as fixed but important biases, 2) they act as context-aware biases, 3) they act as context-aware scaling factors to the attention mechanism. The author trained GPT-2 with different attention variants to verify these hypotheses. The experimental results suggest that the outliers should be acting as implicit context-aware scaling factors. The reasoning process of the paper is clear and convincing. The self-attention with explicit context-aware scaling factor can also stabilize LLM training.

**Weaknesses:**

The author claims that the work "deepens the theoretical understanding" of outliers in LLMs in the Conclusion section. However, there is no theory involved in the analysis and the finding is mostly empirical. On the other hand, the author only conducted experiments with GPT-2. Since the Llama architecture is not exactly the same as GPT-2, the author can also verify the finding with small-scale Llama model.

**Questions:**

Have you tried changing the self-attention in Llama to the "explicit context-aware scaling factor" variant? Will it also remove the outliers?

---

> ### Author Response · Authors · 2024-11-19
> **Official Comment by Authors**
>
> Thank you for your insightful summary and positive feedback on our analysis of outliers in LLMs. We are especially grateful for your recognition of our hypotheses and the clear reasoning behind our experiments, as well as your appreciation of the proposed context-aware scaling mechanism. Your encouraging words motivate us to further refine and extend our work.

---

> > ### Author Response · Authors · 2024-11-19
> > **Verifying Findings on Llama Models**
> >
> > Thank you for your question and suggestion. Due to resource and time constraints, we conducted a brief verification by training a TinyLlama-120M model using the training code referenced from [TinyLlama](https://github.com/keeeeenw/TinyLlama/blob/main/pretrain/tinyllama.py). The results were consistent with those observed for GPT-2: when the self-attention mechanism was replaced with the "explicit context-aware scaling factor" variant, the outliers disappeared.
> >
> > This finding aligns with the theoretical analysis discussed earlier. In standard attention mechanisms, achieving $\text{MHA}(x) \approx 0$ updates during training generates extremely large gradients because of the softmax in self-attention, leading to the formation of outliers in activations and weights. In contrast, the explicit context-aware scaling factor achieves the same objective without producing large gradients, thereby avoiding outlier formation.

---

> > > ### Author Response · Authors · 2024-11-21
> > > **Clarifying the Theoretical Contributions of the Work**
> > >
> > > Thank you for pointing out the concern regarding the theoretical contributions of our work. To address this, we have provided a detailed mathematical framework to analyze the role of **softmax in the formation of systematic outliers** in transformer models, which can be found in [this comment](https://openreview.net/forum?id=rLX7Vyyzus&noteId=SdpyWxpXjJ). Additionally, we have introduced **rigorous mathematical definitions** of outliers and Overlap, which are critical to understanding their impact on model behavior, as outlined in [this response](https://openreview.net/forum?id=rLX7Vyyzus&noteId=bsQeXfU7Qj) and [this one](https://openreview.net/forum?id=rLX7Vyyzus&noteId=00vWrbUPP7). These theoretical analyses complement the empirical findings.

---

> > > > ### Comment · Reviewer_hfXR · 2024-11-30
> > > >
> > > > The author's rebuttal has addressed my concerns and I'll increase the score.

---

> > > > > ### Author Response · Authors · 2024-12-01
> > > > > **Official Comment by Authors**
> > > > >
> > > > > Thank you for your thoughtful feedback and for taking the time to review our rebuttal. We sincerely appreciate your acknowledgment of our efforts to address your concerns and are grateful for your positive reassessment of our work.

---

### Official Review · Reviewer_85qx · 2024-11-07

**Soundness:** 3
**Presentation:** 4
**Contribution:** 3
**Rating:** 8
**Confidence:** 3

**Summary:**

This paper presents a systemic analysis of outliers in transformer language models, which as extremely large values in weights and or activations. Through a set of experiments they show how different types of activations are connected to each other, and how they are all connected to attention scores in the end. In particular, the authors propose outlier activations are used to scale attention activations and allow for a 'zero' update when there is no need to update the residual stream. They verify this hypothesis by training transformer models with different transformer setups and show that including a specific scale parameter removes the existence of outliers from the model.

**Strengths:**

- Very well written. Presents a nice and easy to follow story of a complex topic. Good figures.
- Good experiments to analyze the situation and support their hypothesis well
- Provides a good explanation for a mysterious and sometimes troublesome behavior observed in transformer models.
- Proposes a (few) modified architectures that solve this problem

**Weaknesses:**

- Notation is a little different from what I'm used to. I think it would be more clear to refer to MLP-layer, Attention layer and residual stream in Figure 4. In particular, down projection input is discussed early on in the paper without explaining what it is, would be useful to at least refer to Fig.4 when its first mentioned.
- Could use some more experimental details at least in the appendix to explain the experimental setups etc in more detail.

**Questions:**

Couldn't the model alternatively learn to do ~0 updates by outputting a value matrix V with all the values small? Do you have any hypothesis why current models instead learn to use attention with an outlier connected to a small vector in V instead of making all vectors in V small?

---

> ### Author Response · Authors · 2024-11-19
> **Official Comment by Authors**
>
> We sincerely appreciate your positive feedback on our clear presentation, thorough experiments, and proposed solutions to a challenging behavior in transformer models. Your recognition of our work's clarity and contributions is highly encouraging.

---

> > ### Author Response · Authors · 2024-11-19
> > **Clarifying Notation and Improving Figure References**
> >
> > Thank you for your excellent suggestion. We will add additional labels to highlight the MLP-layer, Attention layer, and residual stream in Figure 4 to enhance its readability. Additionally, we will refer to Figure 4 when the "down projection input" is first mentioned (in Figure 1’s caption) to assist readers in understanding the concept.

---

> > > ### Author Response · Authors · 2024-11-19
> > > **Enhancing Experimental Details in the Appendix**
> > >
> > > Thank you for your suggestion. We completely agree and will expand the appendix with a section detailing the experimental setups. This will include visualizations of existence and position, dimensional consistency analysis, training setups for attention variants on GPT-2, hyperparameters for convergence experiments, and information on inputs and datasets. We hope this will make the experiments easier to understand and replicate.

---

> > > > ### Author Response · Authors · 2024-11-19
> > > > **On the Choice of Outlier Values in Multi-Head Attention for ~0 Updates**
> > > >
> > > > Thank you for this insightful question! To achieve $\text{MHA}(x) \approx 0$ updates, three potential mechanisms can be hypothesized:
> > > >
> > > > - All values $V$ small, independent of key-query interactions:
> > > >   This implies $V = \mathbf{0}$ or $V \approx \mathbf{0}$, rendering the attention update negligible regardless of the inner products $QK^T$.
> > > >
> > > > - Value $v$ corresponding to the current token small, with its associated key and query interaction dominating:
> > > >   Here, the key $k_{\text{current}}$ and query $q_{\text{current}}$ produce large inner products, while $v_{\text{current}}$ is small.
> > > >
> > > > - A fixed outlier token with a small value $v_{\text{outlier}}$, whose key $k_{\text{outlier}}$ interacts strongly with the current token’s query $q_{\text{current}}$:
> > > >   The strong inner product $q_{\text{current}} \cdot k_{\text{outlier}}^T$ ensures the attention mechanism focuses on this outlier, while the small $v_{\text{outlier}}$ enforces a negligible update.
> > > >
> > > > ### Evaluating the Mechanisms
> > > >
> > > > - All values small ($V \approx \mathbf{0}$):
> > > >   This approach is not feasible because $K$ and $V$ from historical tokens are independent of the current token. If all values are small, all subsequent $\text{MHA}(x)$ computations lose access to contextual information encoded in historical keys. Hence, this disrupts meaningful information flow.
> > > >
> > > > - Current token’s value small ($v_{\text{current}} \approx \mathbf{0}$):
> > > >   Similarly, if the value of the current token is always small, future tokens will not be able to extract meaningful context from it. This would degrade the model’s ability to utilize past information for downstream tokens.
> > > >
> > > > - Outlier mechanism:
> > > >   By introducing an outlier token with a weak semantic role and a small $v_{\text{outlier}}$, the model can selectively enforce $\text{MHA}(x) \approx 0$ when the current query $q_{\text{current}}$ strongly aligns with $k_{\text{outlier}}$. Importantly, this mechanism is adaptive, allowing selective updates, and does not interfere with other tokens retrieving contextual information.
> > > >
> > > > ### Key Insights
> > > >
> > > > These mechanisms suggest two essential requirements for achieving $\sim 0$ updates effectively:
> > > >
> > > > - Adaptivity: The mechanism must allow selective enforcement of $\sim 0$ updates as needed.
> > > > - Non-disruption: The mechanism should not prevent future tokens from accessing contextual information.
> > > >
> > > > While these three mechanisms do not exhaust all possibilities, they illustrate the essential conditions for achieving $\sim 0$ updates. Among the proposed mechanisms, only the Outlier Mechanism meets these criteria, making it a plausible explanation for the observed behavior.

---

> > > > > ### Comment · Reviewer_85qx · 2024-12-02
> > > > > **Response to Rebuttal**
> > > > >
> > > > > Thank you for the response! This addresses the questions I had in the review.
> > > > >
> > > > > However, after reading the other reviews and more related work, I have a new concern related to the novelty this approach, in particular the similarity with [1] and whether that is fairly represented in the current manuscript. In particular, I'm concerned with Section 5, where first reading made it seem like the approaches that fix this problem (d and e in Table 2) are original contributions of this paper, while previous work suggested approaches that don't work i.e. (b). However this is not the case and a working approach (d) was directly proposed by [1]. I think at minimum this need to be made more clear in the section, for example by adding citation to [1] on the row for (d) in Table 2. Can you comment on this and the differences between your work and [1]?
> > > > >
> > > > >  [1] Sun, Mingjie, et al. "Massive activations in large language models." arXiv preprint arXiv:2402.17762 (2024).

---

> > > > > > ### Author Response · Authors · 2024-12-03
> > > > > > **Response to Reviewer Comment on Novelty**
> > > > > >
> > > > > > Thank you for raising concerns about the novelty of our approach and its connection to Sun et al. [1]. We appreciate this opportunity to clarify the distinctions and provide a more detailed explanation of our contributions.
> > > > > >
> > > > > > In the revised manuscript, we have already cited Sun et al. in line 375 to acknowledge the connection of one of the hypotheses to their work. In response to your suggestion, we will also explicitly include a citation to [1] in Table 2, row (d), to provide additional clarity.
> > > > > >
> > > > > > ### Technical Advancements Over Sun et al.
> > > > > >
> > > > > > While Sun et al. propose the "Fixed but Important Biases" hypothesis and suggest (d) Attention Bias as a solution, our work introduces several novel contributions that go beyond their scope:
> > > > > >
> > > > > > - **Unified Framework for Outliers:**
> > > > > >    We offer a comprehensive definition of different types of outliers and analyze their relationships and lifecycles, providing a systematic framework for understanding their dynamics within large language models (LLMs).
> > > > > >
> > > > > > - **Challenging Existing Hypotheses:**
> > > > > >    We critically evaluate and experimentally demonstrate the limitations of the "Fixed but Important Biases" hypothesis proposed by Sun et al. Our findings reveal that their explanation does not fully capture the nature and role of outliers in self-attention.
> > > > > >
> > > > > > - **New Interpretation as Context-Aware Scaling Factors:**
> > > > > >    We propose a novel interpretation of outliers, framing them as **context-aware scaling factors** that emerge naturally from the softmax operation in self-attention. This perspective shifts the focus from static biases to dynamic, context-dependent behavior.
> > > > > >
> > > > > > - **Proposal of Explicit Context-Aware Scaling Factor (e):**
> > > > > >    Building on this new interpretation, we introduce (e) Explicit Context-Aware Scaling Factor, which effectively decouples biases and scaling factors. This approach not only resolves outliers more effectively but also provides deeper insights into their functional role within LLMs.
> > > > > >
> > > > > > ### Summary of Contributions
> > > > > >
> > > > > > These advancements mark a significant step forward in understanding and addressing outliers in LLMs, combining theoretical insights with practical implications. We believe that our work complements and extends the findings of Sun et al., offering a refined perspective and innovative solutions to this critical area of research.
> > > > > >
> > > > > > We will incorporate these distinctions more explicitly in the revised manuscript to address your feedback and further clarify the originality of our contributions.
> > > > > >
> > > > > > ---
> > > > > >
> > > > > > **References**
> > > > > > [1] Sun, Mingjie, et al. "Massive activations in large language models." *arXiv preprint arXiv:2402.17762* (2024).

---

> > > > > > > ### Comment · Reviewer_85qx · 2024-12-03
> > > > > > >
> > > > > > > Thank you for the response. I will retain my original rating, even though I think the contributions are a little weaker than I initially thought, and I would probably rate the paper as a 7 if that was an option. I still recommend acceptance as I think the new findings are important and interesting but need to presented in the correct light and in particular make it more clear in the paper what is your contribution vs the contribution of [1].

---

> > > > > > > > ### Author Response · Authors · 2024-12-03
> > > > > > > > **Official Comment by Authors**
> > > > > > > >
> > > > > > > > Thank you for your thoughtful feedback and for recognizing the importance of our findings. We appreciate your constructive comments and will work to ensure our contributions are clearly distinguished and presented appropriately.

---

### Comment · Area_Chair_8Q8q · 2024-11-25

Dear reviewers,

As the deadline for discussion is ending soon. Please respond to the authors to indicate you have read their rebuttal. If you have more questions, now is the time to ask. This is important since the paper is currently undergoing extremely divergent scores.

AC

---

### Author Response · Authors · 2024-11-26
**General Response**

We sincerely thank Reviewer 85qx, Reviewer hfXR, Reviewer pdDU, Reviewer 1brf, Reviewer YjoR, and Area Chair 8Q8q for their valuable feedback and thoughtful suggestions. Their insights have significantly contributed to improving the quality and clarity of our work. Below, we summarize the major revisions made in response to their comments:

1. **Supplementing Experimental Settings**
   Detailed descriptions of all experimental setups have been added to Appendix B to ensure transparency and reproducibility. [Reviewer 85qx, Appendix B]

2. **Improving Readability and Refining Figures**
   We have revised the manuscript to enhance readability, adjusted the placement of figures, and clarified their details for better interpretation. [Reviewer 85qx, Reviewer pdDU, Reviewer YjoR]

3. **Adding Outlier Elimination Experiments in LLaMA Architecture**
   A new experiment demonstrates the effectiveness of outlier elimination in the LLaMA architecture, further validating the generalizability of our approach. [Reviewer hfXR, Appendix D.2]

4. **Providing Rigorous Mathematical Definitions of Outliers and Consistency**
   We have added precise mathematical definitions and calculations for outliers and their consistency to strengthen the theoretical foundation. [Reviewer pdDU, Reviewer YjoR, Section 3 and Appendix B.2]

5. **Extending Mathematical Analysis of Outlier Formation**
   An expanded mathematical analysis in Appendix C provides deeper insights into how softmax causes systematic outliers, addressing key questions raised by reviewers. [Reviewer pdDU, Appendix C]

6. **Including Quantitative Analysis of Memory Consumption and Compression Improvements**
   Additional experiments quantify memory usage and demonstrate the effectiveness of our approach in improving compression efficiency. [Reviewer pdDU, Reviewer 1brf, Section 5]

7. **Analyzing the Impact of Sequence Length on Outliers**
   We analyze how sequence length influences the positioning and occurrence of outliers, adding new insights in Appendix D.3. [Reviewer pdDU, Appendix D.3]

8. **Enhancing Convergence Experiments**
   Training convergence experiments have been expanded to include longer training steps, illustrating the behavior of different attention mechanisms. [Reviewer YjoR, Figure 12]

We believe these revisions substantially address the reviewers' comments and enhance the contributions of our work. We look forward to further discussion and welcome additional feedback to continue refining this study.

---

### Comment · Area_Chair_8Q8q · 2024-12-03

Dear all reviewers,

Thanks for your efforts in engaging with the authors during the rebuttal phase. As it comes to an end, I want to personally remind **Reviewer YjoR** and **pdDU** to further engage with the authors:

- *Reviewer YjoR*: you claimed to increase the score if a new revision of the paper is provided with sufficient rebuttal. Please respond to the authors since they have now a new version with more detailed rebuttal.
- *Reviewer pdDU*: please do repond to authors as they have made considerable effort to reply to you.

Thanks,

AC

---

> ### Comment · Reviewer_YjoR · 2024-12-03
>
> Thanks, updated. The rebuttal and revision are well done.

---

### Meta-Review · Area_Chair_8Q8q · 2024-12-17

**Metareview:**

This paper presents a systematic investigation of outliers in LLMs, offering a novel perspective on the intricate relationships between activation, weight, and attention outliers across various LLMs (85qx, hfXR). With most reviewers rating the soundness and contribution as "good" (3 out of 5), the study provides a compelling empirical analysis that deepens our understanding of model behavior (reviewers 85qx, hfXR). The research offers significant strengths, including comprehensive visualization of outlier patterns, a detailed exploration of their lifecycle, and an innovative approach to addressing these systematic outliers (pdDU). Notably, the authors propose a context-aware scaling mechanism that demonstrates potential improvements in model convergence, with Figure 15 presenting an intriguing result showing the possibility of converging to a lower loss at a faster rate (YjoR).

Despite some reservations about the primarily empirical nature of the work, reviewers acknowledged the paper's important contributions to understanding LLM internals (1brf, pdDU). The key weaknesses identified include limited comparison with existing methods, and the need for more extensive experimental validation (1brf, YjoR). Reviewers suggested areas for improvement, such as providing clearer definitions of outliers, expanding the investigation across more recent LLM architectures, and conducting more comprehensive convergence studies (YjoR, pdDU). However, the overall consensus leans towards acceptance, with reviewers appreciating the paper's systematic approach to investigating outliers and its potential implications for model optimization and understanding (85qx, hfXR). The study's strength lies in its comprehensive visualization, detailed analysis of outlier patterns, and the proposed method for addressing these systematic outliers in transformer models.

**Additional Comments On Reviewer Discussion:**

Reviewer pdDU questioned the paper for only adopting "outdated" models (i.e., Llama2) for experiments as not sufficient. However, reviewer 1brf expressed his support for the paper and claimed the model is not a big issue. As the AC, I agree with 1brf since Llama2 is not outdated and this is a research paper where we should pay attention to the technical contributions rather than base models. The method is general and can be applied to any models.

---

### Decision · Program_Chairs · 2025-01-22

Accept (Poster)